# Testing the near-field Gaussian plume inversion flux quantification technique using unmanned aerial vehicle sampling

Adil Shah[1], Joseph R. Pitt[1], Hugo Ricketts[1, 2], J. Brian Leen[3], Paul I. Williams[1, 2], Khristopher Kabbabe[4], Martin W. Gallagher[1], Grant Allen[1]

[1]Centre for Atmospheric Science, The University of Manchester, Oxford Road, Manchester, M13 9PL, United Kingdom
[2]National Centre for Atmospheric Science, The University of Manchester, Oxford Road, Manchester, M13 9PL, United Kingdom
[3]ABB – Los Gatos Research, 3055 Orchard Drive, San Jose, CA 95134, California, United States of America
[4]School of Mechanical, Aerospace and Civil Engineering, The University of Manchester, Oxford Road, Manchester, M13 9PL, United Kingdom

*Correspondence to*: Adil Shah (adil.shah@manchester.ac.uk)

**Abstract.** Methane emission fluxes from many facility-scale sources may be poorly quantified, potentially leading to uncertainties in the global methane budget. Accurate atmospheric measurement based flux quantification is urgently required to address this. This paper describes the first test (using unbiased sampling) of a near-field Gaussian plume inversion (NGI) technique, suitable for facility-scale flux quantification, using a controlled release of methane gas. Two unmanned aerial vehicle (UAV) platforms were used to perform 22 flight surveys downwind of a point-source methane gas release from a regulated and flow-metered cylinder. One UAV was tethered to an instrument on the ground, while the other UAV carried an on-board prototype instrument, both of which used the same near-infrared laser technology. Both instruments were calibrated using certified standards, to account for variability in the instrumental gain factor, assuming fixed temperature and pressure. Furthermore, a water vapour correction factor, specifically calculated for the instrument, was applied and is described here in detail. We also provide guidance on potential systematic uncertainties associated with temperature and pressure, which may require further characterisation for improved measurement accuracy. The NGI technique was then used to derive emission fluxes for each UAV flight survey. We found good agreement of most NGI fluxes with the known controlled emission flux, within uncertainty, verifying the flux quantification methodology. The lower and upper NGI flux uncertainty bounds were, on average, 17%±10(1$\sigma$)% and 227%±98(1$\sigma$)% of the controlled emission flux, respectively. This range of conservative uncertainty bounds incorporate factors including the variability in the position of the time-invariant plume and potential for under-sampling. While these average uncertainties are large compared to methods such as tracer dispersion, we suggest that UAV sampling can be highly complementary to a toolkit of flux quantification approaches and may be a valuable alternative in situations where site access for tracer release is problematic. We see a tracer release combined with UAV sampling as an effective approach in future flux quantification studies. Successful flux quantification using the UAV sampling methodology described here demonstrates its future utility in identifying and quantifying emissions from methane sources such as oil and gas extraction infrastructure facilities, livestock agriculture and landfill sites, where site access may be difficult.

## 1 Introduction

Methane is the second most important anthropogenic greenhouse gas (Etminan et al., 2016), with an important role in atmospheric chemistry processes (Ehhalt et al., 1972). There is more methane in the atmosphere today than there has even been over the past 800 000 years (Etheridge et al., 1998; Loulergue et al., 2008; Earth System Research Laboratory, 2020). The global methane budget is subject to significant uncertainties (Kirschke et al., 2013; Saunois et al., 2016b; Nisbet et al., 2019), particularly from inventory uncertainty in facility scale sources such as landfill sites (Scheutz et al., 2009), herds of cattle (Blaxter and Clapperton, 1965) and oil and gas extraction infrastructure (Brantley et al., 2014), which collectively

contribute significantly to global methane emissions (Dlugokencky et al., 2011; Saunois et al., 2016a). These uncertainties can be reduced through the accurate source identification and subsequent quantification of methane emission fluxes using top-down (atmospheric measurements based) methods, in order to validate bottom-up (inventory based) emission flux estimates (Lowry et al., 2001; Nisbet and Weiss, 2010; Allen, 2016; Desjardins et al., 2018).

45

Accurate top-down flux quantification from facility scale sources requires a combination of wind vector measurements along with in situ measurements of atmospheric methane mole fraction (Dlugokencky et al., 1994; Rigby et al., 2017). Facility-scale emission fluxes can be derived from near-field sampling (less than 500 m from the source), which may be acquired from an unmanned aerial vehicle (UAV) platform (Gottwald and Tedders, 1985). UAVs are cheap, versatile and relatively easy to use (Villa et al., 2016), compared to large manned aircraft (Illingworth et al., 2014; Lehmann et al., 2016). They can fly near to source and can be directed automatically using waypoints, to enable even and unbiased spatial sampling (Greatwood et al., 2017; Feitz et al., 2018). There are three principal approaches for measuring methane mole fraction from a UAV in situ: on-board air samples can be collected for subsequent analysis (Chang et al., 2016; Greatwood et al., 2017; Andersen et al., 2018), air can be pumped through a long tube to a sensor on the ground for analysis (Brosy et al., 2017; Wolf et al., 2017; Shah et al., 2019) or air can be analysed live using a sensor mounted on-board the UAV (Berman et al., 2012; Khan et al., 2012; Nathan et al., 2015; Golston et al., 2017; Martinez et al., 2020). Yet, a key limitation to accurate source identification and flux quantification is the precision and accuracy of methane mole fraction measurements (Hodgkinson and Tatam, 2013). Miniaturised sensors suitable for UAV sampling are emerging (Villa *et al.*, 2016), but high precision lightweight in situ closed path sensors, featuring superior techniques, such as off-axis integrated cavity output spectroscopy, have not yet materialised.

Some studies have used UAV remote sensing measurements to derive emission fluxes (Golston et al., 2018; Yang et al., 2018). However, to our knowledge, only Nathan et al. (2015) have derived fugitive methane emission fluxes using UAV in situ measurements. In that study, a UAV with an on-board in-situ low precision sensor ($\pm0.1$ ppm at 1 Hz) flew in orbits around a gas compressor station, using mass balance box modelling, with geospatial kriging for interpolation, to derive the emission flux. However this method was not tested for UAV sampling with an accurate known (controlled) methane flux rate. It is crucial that novel flux quantification techniques are tested by sampling a known flux, prior to investigating unknown emission sources (Desjardins et al., 2018; Feitz et al., 2018). Our previous study, was the first test of an in situ flux quantification technique using UAV sampling downwind of a controlled methane release, where a UAV was connected to a high precision methane analyser on the ground using 150 m of tubing (Shah et al., 2019). A data-set of two-dimensional downwind sampling measurements, on a vertical flux plane, was used to develop the near-field Gaussian plume inversion (NGI) technique for flux quantification, as other flux quantification approaches failed (Shah et al., 2019). Fully manual UAV piloting was employed in this previous study to actively pursue the position of the time-invariant emission plume on the sampling plane, using mid-flight knowledge of its position. This resulted in calculated emission fluxes that were significantly positively biased compared to known emission fluxes; this represents a source of vulnerability in fully manual UAV sampling, which we address in this work.

Here we test the application of the NGI method with unbiased UAV sampling of controlled methane emission sources, by flying two UAVs downwind of the release. In this work, the causes of positive flux bias reported in Shah et al. (2019) were addressed in our sampling strategy, by flying a UAV without prior knowledge of the position of the time-invariant emission plume. One UAV was connected to a commercially available instrument on the ground and the other carried a lighter prototype on-board instrument (sect. 3). Both instruments were characterised and calibrated, with the effects of cell pressure and cell temperature also assessed (sect. 2). Our approach to water vapour correction is also outlined in sect. 2. Limitations

to our sensor characterisation procedures and future improvements are also outlined. Sampled data was then used to derive NGI flux uncertainty ranges (sect. 4) for each of 22 flight surveys. In sect. 5 the success of the NGI method is assessed overall and its sampling constraints are summarised.

## 2 Methane instrumentation and calibration

### 2.1 Instrumental overview

Two instruments were used to derive atmospheric dry methane mole fraction ($[X]$) measurements during UAV sampling. $[X]$ is given in units of parts-per-million (ppm) throughout this paper, which are defined here as the number of moles of methane per million moles of dry air ($10^{-6} \cdot mol_{methane} \, mol^{-1}$), with parts-per-million (ppb) defined as the number of moles of methane per billion moles of dry air ($10^{-9} \cdot mol_{methane} \, mol^{-1}$). In this section, the ABB Micro-portable Greenhouse Gas Analyzer (MGGA) and a lighter prototype MGGA (pMGGA), designed for UAV use, are compared and characterised to assess their performance, albeit under ambient (variable) laboratory temperature and pressure conditions. The technical specifications of both instruments are compared in Table 1. Both instruments use off-axis integrated cavity output spectroscopy (ICOS) to derive simultaneous empirical measurements of methane, carbon dioxide and water mole fraction, from the absorption of a near-IR (1651 nm) laser, with the water and methane absorption peaks separated by 0.2 nm. The pMGGA uses an additional laser (1603 nm) to measure carbon dioxide mole fraction more accurately. Off-axis ICOS techniques reflect a tuneable laser between two mirrors in a high-finesse optical cavity, to obtain high-precision mole fraction measurements (see Paul et al. (2001) and Baer et al. (2002) for further details on off-axis ICOS).

The e-folding time of the high-finesse cavity in both sensors was measured here by fitting an exponential decay function to the transition from a high to low mole fraction standard gas (see Table 1 for results, with sensor flow rate also given). This represents the time taken for 63.2% of the contents of the high-finesse cavity to be replaced. The Allan variance of each sensor was also derived (see Fig. 1 and Fig. 2), by sampling a dry gas standard continuously (17 hours and 23 minutes for the MGGA and 38 hours and 30 minutes for the pMGGA), under ambient conditions. The 1 Hz and 0.1 Hz Allan deviation for both instruments is given in Table 1. The sampling noise uncertainty ($\sigma_n$), used within the total mole fraction enhancement uncertainty (discussed in sect. 2.4), represents the Allan deviation at the maximum sampling frequency. $\sigma_n$ for the MGGA and pMGGA are 2.71 ppb (at 10 Hz) and 5.44 ppb (at 5 Hz), respectively. The optimum Allan variance integration time was also assessed for each sensor ($(20\pm3)$ s for the MGGA and $(70\pm10)$ s for the pMGGA); this represents maximum sampling time before instrumental drift begins to dominate over instrumental noise. During the MGGA Allan variance test, cell temperature (which varied between 24.9° C and 27.8° C) and cell pressure (which varied between 1.0093 bar and 1.0128 bar) were also recorded to assess their correlation with $[X]$ (see Fig. S1 and Fig. S2). Correlation of both cell temperature and cell pressure was poor, with Pearson correlation coefficients of -0.4849 and -0.3835, respectively, and linear gradients of -0.0022 ppm° $C^{-1}$ and -0.0022 ppm mbar$^{-1}$, respectively. Thus over a limited cell pressure and cell temperature range, there was no definitive correlation with $[X]$ for the MGGA under typical laboratory conditions, though there may be a need for a more comprehensive cell temperature and cell pressure characterisation in the future, depending on the expected sampling conditions.

### 2.2 Empirical water vapour correction

Raw wet methane mole fraction measurements ($[X]_0$) recorded by each instrument were corrected for influence of atmospheric water vapour on mole fraction retrievals. Water vapour influences measurements of dry methane mole fraction ($[X]$) for three main reasons (Karion et al., 2013; O'Shea et al., 2013; Rella et al., 2013). First and most significantly, dilution

effects occur, where the bulk presence of water reduces the quantity of methane in the cavity at a given pressure. Second, strong, broad infrared absorption bands of water can interfere with the absorption spectrum of methane, though this effect is thought to be small in this case as the spectral lines are well separated in the spectral sampling region of these instruments. Third, pressure broadening can alter the shape of the methane spectral absorption band, due to collisional interaction between water and methane molecules, compared to pressure broadening without water in the cavity. The combined impact of pressure broadening absorption band changes and dilution has a net effect of decreasing $[X]_0$ in both instruments, based on laboratory observations at a range of methane and water mole fractions, under typical near-surface conditions.

To account for pressure broadening absorption band changes, both the MGGA and pMGGA use an internal retrieval algorithm to derive methane mole fractions, which includes empirically derived estimates of the effect of pressure broadening as a function of varying empirical water vapour mole fraction. The instruments then output raw dry mole fraction measurements, which have additionally been corrected for the effect of mole fraction dilution by water vapour, and raw wet methane mole fraction measurements ($[X]_0$), which have not been corrected for dilution (but are still calculated using the same empirically derived pressure broadening correction, as a function of water mole fraction). A typical pressure broadening correction (as a function of water mole fraction) is determined by the manufacturer based on experiments conducted with a sample batch of instruments, yielding an average correction applied to all instruments. However, because the correction convolves pressure broadening absorption band changes due to water vapour with pressure broadening absorption band changes due to instrument factors, there is some variability from unit to unit. Therefore, to obtain a more accurate correction for the influence of water vapour on the individual instruments used here, we apply a further empirical post-processing correction factor to $[X]_0$ measurements (without the dilution correction) reported by the instruments under ambient laboratory conditions, using reported measurements of water mole fraction ($[H_2O]$). Although $[H_2O]$ measurements reported by the instruments may not be an accurate representation of the true water mole fraction in the cavity, they are sufficient for an empirical correction on $[X]_0$, provided that $[H_2O]$ does not drift and is independent of dry uncalibrated methane mole fraction ($[X]_0^{dry}$). Therefore $[H_2O]$ was not calibrated against standards and an instrumental reported value was used for this empirical correction.

For the water correction to be valid, $[H_2O]$ should be independent of $[X]_0^{dry}$. However both instruments reported small but non-zero $[H_2O]$ when sampling dry air, which decreased with increasing $[X]_0^{dry}$. Therefore before a water correction could be applied, a $[H_2O]$ baseline ($[H_2O]_0$) was derived under ambient (variable) laboratory conditions up to 5 ppm, which represents the upper limit of the World Meteorological Organisation Greenhouse Gas Scale (WMO-X2004A) for methane. Gas from two cylinders with different methane compositions (1.901 ppm and 5.049 ppm) was dried by passing it through a water trap (a stainless-steel coil immersed in solid carbon dioxide pellets) before being sampled by both the MGGA and the pMGGA. Dry air from an additional cylinder (2.167 ppm) was also sampled by the MGGA. Each gas was sampled a minimum of 11 times for 4-minute periods, from which 1-minute averages were taken. $[H_2O]_0$ decreased with $[X]_0^{dry}$, given by Eq. (1), where $a$ is the water baseline offset and $b$ is the water baseline coefficient. The data used to fit $[H_2O]_0$ is plotted in Fig. S3 and Fig. S4.

(1)     $[H_2O]_0 = a + (b \cdot [X]_0^{dry})$

$a$ and $b$ for both instruments are given in Table 2. The effect of $[H_2O]_0$ changes on $[X]_0^{dry}$ beyond the 5 ppm range was also tested up to approximately 100 ppm (see SI for details).

$[H_2O]_0$ is assumed here to be relatively constant over time. To test this, an Allan variance test was conducted on $[H_2O]_0$ for both instruments (see Fig. S5 and Fig. S6), using the same Allan variance data-set described in the previous section. This revealed a water baseline Allan deviation precision for the MGGA and pMGGA of $\pm 16 \cdot 10^{-6}$ mol$_{water}$ mol$^{-1}$ and $\pm 27 \cdot 10^{-}$

$^6$ mol$_{water}$ mol$^{-1}$, respectively, using a 1-minute integration time (the averaging time used for each [H$_2$O]$_0$ point). These 1-minute Allan deviation averages are small compared to the water vapour content of typical tropospheric air, suggesting that [H$_2$O]$_0$ remains relatively stable. Having established a stable and well characterised water baseline (assuming ambient temperature and pressure conditions), a post-processing empirical water correction factor ($v$) was derived by sampling gas from a single cylinder (2.205 ppm for the MGGA and 2.183 ppm for the pMGGA), which was humidified to 9 fixed dew points (from 0 °C to 18 °C), using a dew point generator (LI-610, LI-COR, Inc.), following a similar experimental set-up used by O'Shea et al. (2013). The humidified gas was first sampled dry (to measure [$X$]$_0^{dry}$), by passing it through the water trap, and then sampled wet (to measure [$X$]$_0$ as a function of [H$_2$O]). An example of sampled [$X$]$_0$ and [H$_2$O] measurements, used to calculate each data point, is given in Fig. S8. A single gas standard was deemed sufficient for this test as both dilution and pressure broadening absorption band changes affect the gain factor on methane mole fraction measurements (i.e. they do not affect the instrumental methane offset). Thus this water correction is assumed to be independent of [$X$]$_0^{dry}$ and solely dependent on the amount of water in the cavity. However any water correction may be systematically influenced by cell temperature and cell pressure. This was not discussed in detail in this work as these effects are deemed to be small under typical near-surface environmental changes compared to the large methane elevations that were measured (see SI for further discussion).

[$X$]$_0$ is then corrected by dividing it by $v$, as $v$ is effectively the ratio between [$X$]$_0$ and [$X$]$_0^{dry}$, as a function of [H$_2$O]. The ratio of [$X$]$_0$ to [$X$]$_0^{dry}$ was plotted against ([H$_2$O] - [H$_2$O]$_0$), where [H$_2$O]$_0$ was the water baseline measured during dry sampling (see Fig. S9 and Fig. S10). Subtracting the baseline in this analysis minimised the effects of [$X$]$_0^{dry}$ on [H$_2$O]. A quadratic fit was applied to both curves, with the intercept forced to unity. The first order coefficient ($\alpha$) and second order coefficient ($\beta$) of the quadratic fit, given in Table 2, were then be used to derive $v$ using Eq. (2), as a function of [H$_2$O].

(2)    $v = 1 + (\alpha \cdot ([H_2O] - [H_2O]_0)) + (\beta \cdot ([H_2O] - [H_2O]_0)^2)$

As [H$_2$O]$_0$ in Eq. (2) is typically unknown, [H$_2$O]$_0$ defined in Eq. (1) can be substituted into Eq. (2), to yield Eq. (3).

(3)    $v = 1 + (\alpha \cdot ([H_2O] - a - (b \cdot [X]_0^{dry}))) + (\beta \cdot ([H_2O] - a - (b \cdot [X]_0^{dry}))^2)$

As [$X$]$_0^{dry}$ in Eq. (3) is also unknown, an approximation that [$X$]$_0^{dry}$ is close to [$X$]$_0$, in typical tropospheric humidity conditions, can be used. Thus Eq. (3) can be rewritten in terms of [$X$]$_0$ and [H$_2$O], using Eq. (4).

(4)    $v \approx 1 + (\alpha \cdot ([H_2O] - a - (b \cdot [X]_0))) + (\beta \cdot ([H_2O] - a - (b \cdot [X]_0))^2)$

To check the above assumption, as a simple example (for the MGGA), when [H$_2$O] is 0.01 mol$_{water}$ mol$^{-1}$ and [$X$]$_0^{dry}$ is 5 ppm, Eq. (3) yields $v$ of 0.98089 whereas Eq. (4) yields a similar value for $v$ of 0.98092. This small $v$ change supports the use of Eq. (4) as an alternative to Eq. (3), by confirming that [$X$]$_0^{dry}$ is close to [$X$]$_0$ in this simple example.

The fit given by Eq. (4) relies on a reliable water baseline, independent of cell pressure and cell temperature. If the MGGA sampled 5 ppm of dry methane and without a baseline correction, $v$ would be 1.0020, thus representing a methane mole fraction reduction of 0.0098 ppm (at 5 ppm), assuming invariant environmental conditions. However, as Eq. (4) acts to remove this small uncertainty, the residual uncertainty would be very small. In addition, the uncertainty in our empirical water correction fit was quantified using each water correction residual ($R$) from Eq. (2), to derive a water fitting uncertainty factor ($\sigma_v$) for each instrument (see Table 2), using Eq. (5). This $\sigma_v$ uncertainty is the standard deviation of the mean of the residuals and quantifies the quality of our applied water correction fits, where $N$ is the total number of residuals. However there may be additional water correction uncertainty due to effects of cell temperature and cell pressure on $v$, which may be useful to examine further in future work.

(5)    $\sigma_v = \left( \frac{\sum (R^2)}{N} \right)^{\frac{1}{2}}$

Our water correction approach (given by Eq. (2)) is analogous to the approach of previous work using the same spectroscopic technique (O'Shea et al., 2013); this previous work found that the water correction is stable and does not drift. Thus an uncertainty in our water correction fit was deemed to be sufficient to characterise uncertainty empirically. To summarise, this is a purely empirical instrument specific correction to correct for the effects of water vapour in the cavity, valid for the tested water mole fraction range of up to 0.016 $mol_{water}$ $mol^{-1}$. For example our correction (assuming constant temperature and pressure) would increase an MGGA $[X]_0^{dry}$ measurement (at 2 ppm) by +0.27%, at a humidity of 0.001 $mol_{water}$ $mol^{-1}$, and by +1.8%, at a humidity of 0.01 $mol_{water}$ $mol^{-1}$, thus improving measurement accuracy.

### 2.3 Calibration

In order to convert $[X]_0$ into $[X]$, both instruments were calibrated by sampling a low standard methane mole fraction ($[X]_{low}$) of 1.901 ppm and a high standard methane mole fraction ($[X]_{high}$) of 5.049 ppm, both of which were certified WMO-X2004A standards. Each gas was sampled intermittently for 4-minute periods of continuous sampling. The water trap was used throughout each calibration as an extra precaution, to ensure dry gas entered the sensor cavities. One-minute averages from each 4-minute sampling period were taken to derive one value of low $[X]_0^{dry}$ ($[X]_0^{dry}{}_{low}$) and one value of high $[X]_0^{dry}$ ($[X]_0^{dry}{}_{high}$) representative for each 8-minute period. The time increment between each $[X]_0^{dry}{}_{low}$ and $[X]_0^{dry}{}_{high}$ value was then interpolated from 8 minutes to 4 minutes, such that every measured value of $[X]_0^{dry}{}_{low}$ had a corresponding interpolated value of $[X]_0^{dry}{}_{high}$ and vice-versa. Individual measured and interpolated $[X]_0^{dry}{}_{low}$ and $[X]_0^{dry}{}_{high}$ values for both instruments are plotted in Fig. S11 and Fig. S12.

These measured and interpolated averages were used to calculate an average gain factor ($G$) and gain factor uncertainty ($\sigma_G$), from the average and standard deviation, respectively, of a set of at least 24 individual gain factors, calculated using Eq. (6) (Pitt et al., 2016).

$$(6) \qquad \text{gain factor} = \frac{[X]_{high} - [X]_{low}}{[X]_0^{dry}{}_{high} - [X]_0^{dry}{}_{low}}$$

The average offset ($C$) and offset uncertainty ($\sigma_C$) was calculated by taking the average and standard deviation, respectively, of individual offsets, calculated using Eq. (7) and Eq. (8) (Pitt et al., 2016).

$$(7) \qquad \text{low offset} = [X]_{low} - (G \cdot [X]_0^{dry}{}_{low})$$

$$(8) \qquad \text{high offset} = [X]_{high} - (G \cdot [X]_0^{dry}{}_{high})$$

$G$, $\sigma_G$, $C$ and $\sigma_C$ for both instruments are given in Table 3. During these calibrations, the cell temperature of the MGGA and pMGGA were $(31.4\pm0.7)°$ C and $(24.6\pm0.1)°$ C, respectively, and the cell pressure of the MGGA and pMGGA were $(1005.9\pm0.2)$ mbar and $(614.30\pm0.01)$ mbar, respectively.

A key advantage of this calibration procedure is that uncertainty in $G$ is well quantified up to $[X]_{high}$, assuming stable cell temperature and cell pressure. Cell temperature and cell pressure both effect spectral fitting parameters and may consequently have an impact on $G$, though this effect would be smaller for the pMGGA which is pressure controlled. The effect of cell temperature on $G$ is small; this was tested by performing a short term (test) calibration with the MGGA at $(44.08\pm0.02)°$ C, yielding a gain factor of 0.9979 (see SI for details). In addition, the MGGA was calibrated at $(968.7\pm0.3)$ mbar, yielding a gain factor of 0.9967 (see SI for details). These test gain factors are both similar to $G$ (from the main calibration) of $0.9970\pm0.0002$. Furthermore, there was no discernible correlation for both cell temperature and cell pressure during the MGGA Allan variance test (see above), which suggests that $G$ is negligibly insensitive to these parameters, over the limited environmental range for the duration of the Allan variance test, though more comprehensive characterisation of these parameters may be required in future work. Although the pMGGA was not tested in this way, we assume similar behaviour due to identical spectroscopic techniques. Nevertheless, separate in-field calibrations would be

preferable to enhance measurement accuracy, by characterising the effect of variability in cell temperature and cell pressure on $G$. However there are logistical challenges with in-field calibrations, such as the need for calibration gases and the time required to perform calibrations in dynamic atmospheric temperature and pressure conditions. The laboratory calibrations described here required at least three hours of sampling to characterise variability in $G$: this may be impractical in field conditions.

## 2.4 Methane enhancement and uncertainty

The calibration procedures described above show that $G$ is almost equal to 1 and $C$ is almost equal to 0, relative to the atmospheric methane background, for both instruments (see Table 3), under ambient (but not controlled) conditions. This means that both instruments record raw $[X]_0$ measurements with very little systematic error, even when uncalibrated. Thus for most methane measurement purposes, $[X]_0$ may not need to be corrected. However in this work, $G$ was applied to $[X]_0$ for improved accuracy.

$[X]$ can be calculated in ppm using Eq. (9).

$$(9) \qquad [X] = \left( \frac{G}{v} \cdot [X]_0 \right) + C$$

However, during flux calculation, the enhancement in methane mass density ($E$), in kg m$^{-3}$, above some background is required and was calculated here using Eq. (10). The background methane mole fraction ($[X]_b$) and corresponding background uncertainty ($\sigma_b$) can be calculated from a subset of $[X]$ measurements, which can be acquired from out-of-plume sampling (see sect. 3). The molar density of dry air ($\rho$) and the uncertainty in $\rho$ ($\sigma_\rho$), in units of dry mol m$^{-3}$, can be derived from pressure, temperature and humidity measurements. The molar mass of methane ($M$) is fixed at 0.01604 kg mol$_{\text{methane}}^{-1}$.

$$(10) \qquad E = ([X] - [X]_b) \cdot \rho \cdot M$$

To calculate the uncertainty in $E$ ($\sigma_E$), the linearity in the instrument response was characterised up to 5 ppm (i.e. the extent of the WMO-X2004A scale). This was achieved by characterising the MGGA response to five certified WMO-X2004A standards. A linear fit was then applied to measured $[X]$, with residuals used to derive an uncertainty due to non-linearity ($\sigma_L$) of ±2.3 ppb (see SI for further details). We adopt the same non-linearity uncertainty factor for the pMGGA as both instruments use identical spectroscopic techniques. $\sigma_E$ can then be calculated by combining $\sigma_b$ with the precision and accuracy uncertainty components of $[X]$, using Eq. (11). Precision is characterised by $\sigma_n$ and accuracy is characterised by $\sigma_L$, $\sigma_G$ and $\sigma_v$ terms. $\sigma_G$ also incorporates the effects of drifts, as it was derived from a prolonged sampling period over which drifts could develop. However $\sigma_E$ does not incorporate uncertainties due to the potential systematic error of cell temperature and cell pressure variations on $E$, which may manifest themselves as an accuracy term in Eq. (11).

$$(11) \qquad \sigma_E = E \cdot \left( \left( (\sigma_n{}^2 + \sigma_L{}^2 + \sigma_b{}^2) \cdot \left( \frac{\rho \cdot M}{E} \right)^2 \right) + \left( \frac{\sigma_G}{G} \right)^2 + \left( \frac{\sigma_v}{v} \right)^2 + \left( \frac{\sigma_\rho}{\rho} \right)^2 \right)^{\frac{1}{2}}$$

Although $M$ remains constant, $v$ in Eq. (11) changes as a function of $[X]_0$ and $[H_2O]$, for each value of $E$. $\sigma_C$ is not required in Eq. (11) as the offset cancels out in Eq. (10), when substituting in Eq. (9). This is an important advantage of using $E$ rather than $[X]$, in the flux analysis used in the following section.

## 2.5 Future improvements for instrumental characterisation

There are a number of steps that can be taken to better characterise both instruments, to account for the effects of cell temperature and cell pressure on instrumental output. The simple tests at the end of sect 2.3 show that both cell temperature and cell pressure can affect $G$, though this effect is subtle over small variations. The Allan variance tests (see sect. 2.1) also

show that influence of cell temperature and pressure on instrumental output is small. Thus future calibrations should be conducted in a controlled environment. Cell temperature and cell pressure may also effect $[H_2O]_0$ (see SI for details). It would also be useful to characterise $[H_2O]_0$ under a wider range of environmental conditions. Furthermore, $[H_2O]_0$ is assumed to respond linearly to $[X]_0^{dry}$ (up to 5 ppm) based on our sampling of two or three gas standards. This linearity could

be tested in future, under controlled conditions, by sampling more standards. The linearity of instrumental $[X]_0^{dry}$ response to $[X]$ could also be tested by sampling more certified standards to populate the range between 2 ppm and 5 ppm. It may also be useful to sample below 2 ppm to fully characterise linearity, including at 0 ppm (using synthetic zero-air). However as the instruments are designed for atmospheric sampling, it is rare to sample air below the atmospheric ambient mole fraction background (approximately 1.9 ppm at the time of writing). On the other hand, although the WMO-X2004A scale does not

exceed 5 ppm, it would be useful to test the linearity in instrumental response at higher mole fractions, using specialised certified gas mixes. Nevertheless, we are confident that Eq. (11) adequately quantifies uncertainties up to 5 ppm, incorporating terms for both accuracy and precision, assuming relatively constant temperature and pressure. Measurement accuracy may be improved by following the above suggestions, but this is not the focus of this work and our instrumental testing was deemed sufficient for our UAV sampling approach, described in the next section.

**3 Method Testing**

**3.1 Experimental description**

A UAV sampling methodology for source identification and flux quantification was tested in two fields adjacent to a natural gas extraction facility in Little Plumpton (near Wesham), Lancashire, United Kingdom (+53.78785° N, -2.94758° E), prior to any drilling or hydraulic fracturing, over five sampling days in August and September 2018. A map of the field site is

given in Fig. 3. The two adjacent grass fields, in which all UAV sampling took place, belong to a fully operational dairy farm. Methane was released from within the operating site at one of two controlled flux rates ($F_0$), from 0.25 m above ground level (see SI for controlled release details). $F_0$ was undisclosed during flux analysis, prior to the comparisons shown later in this paper, allowing for blind method testing.

Two adapted DJI Spreading Wings S1000+ octocopter UAVs (labelled UAV1 and UAV2) were used to sample the methane plume on a downwind vertical plane, roughly perpendicular to mean wind direction (see Table 4 for UAV details). The location of the UAVs in relation to the controlled release and their sampling paths was decided on each day based on public wind forecasts and on-site wind measurements, to horizontally centre (as best as possible) each UAV flight track downwind of the controlled release. $[X]$ measurements from both platforms are given in Fig. 4. UAV1 was operated using pre-

programmed waypoints and ascended diagonally. Each UAV1 flight survey was composed of two parts: one flight to the right of the source (projected onto the sampling plane, perpendicular to mean wind direction) and one to the left. Meanwhile each UAV2 flight survey was composed of a single flight, to perform horizontal transects, with each transect at a roughly fixed height, up to approximately 100 m laterally away from the take-off position. 7 surveys were conducted by UAV1 (labelled, T1.1 – T1.7) and 15 surveys were conducted by UAV2 (labelled, T2.1 – T2.15). Individual flight survey details are

given in Table S1 and Table S2.

UAV1 (see Fig. 5) was connected to the MGGA on the ground, using 150 m of perfluoroalkoxy (PFA) tubing (4.76 mm inner diameter; 6.35 mm outer diameter). Air was pulled through the tubing using a small pump (NMS 030.1.2 DC-B 12V, KNF Neuberger UK Ltd), from which the MGGA subsampled. The sampling lag time between air entering the UAV air inlet

and air entering the MGGA cavity was 25 s, with an average volumetric flow rate through the tube of $(110 \pm 10)$ cm$^3$ s$^{-1}$ and a

flow rate through the instrument (at ambient pressure) of $(27.90\pm0.05)$ cm$^3$ s$^{-1}$. Both the MGGA and the pump were powered by a 12 V lead-acid battery. As the tether connected to UAV1 occasionally kinked during flight, blocking air through the tube, 16% of all $[X]_0$ sampling from UAV1 was discarded (such periods were identified and recorded in the field from the flow of air to the pump). The pMGGA was mounted on board UAV2 (see Fig. 5), beneath the centre frame. The sampling lag time between air entering the external air inlet and air entering the pMGGA cavity was 2 s, with a flow rate through the instrument of $(5.08\pm0.02)$ cm$^3$ s$^{-1}$. The pMGGA was powered using the on-board 22.2 V UAV2 battery. Both the MGGA and pMGGA transmitted live, real-time mole fraction measurements wirelessly, to a tablet computer on the ground. Satellite geolocation was recorded by the pMGGA, on-board UAV2, simultaneous with every $[X]_0$ measurement. Satellite geolocation was recorded on UAV1 by a separate on-board computer, sampling at 1 Hz. Aerial UAV flight tracks are given in Fig. S14 for UAV1 and Fig. S15 for UAV2.

A lightweight wind sensor (FT205EV, FT Technologies Limited) was mounted on-board UAV1, on a carbon fibre pole 305 mm above the plane of the propellers (see SI for further details and testing). It recorded wind speed and direction at 4 Hz. These measurements were used to model change in wind speed with height above ground level ($z$). A two-dimensional stationary sonic anemometer (WS500-UMB Smart Weather Sensor, G. Lufft Mess- und Regeltechnik GmbH) was also situated on the southern boundary of the operating site (see Fig. 3), $(3.30\pm0.03)$ m above ground level. This provided wind speed, wind direction, relative humidity, temperature and pressure measurements every minute. Wind measurements from both sensors were combined to derive the average absolute wind speed as a function of $z$ ($WS(z)$), for the duration of each flight survey. This is described in detail in the SI.

The position of the UAV1 wind sensor and the positon of the air inlet for both UAVs, relative to the plane of the propellers, are shown in Fig. 5. In hindsight, the UAV2 air inlet should have been elevated above the plane of the propellers, as downwash from the rotating propellers can distort the apparent plume morphology, leading to small errors in the geospatial positioning of the sampled air (Schuyler and Guzman, 2017; Zhou et al., 2018). As UAV2 generally sampled at a greater distance from the emission source than UAV1, allowing the instantaneous plume to disperse across a larger area, the impact of such a geospatial positioning error is expected to the small. Nevertheless care should be taken in future work to reduce these potential sampling biases.

**3.2 Flux density measurements**

Each individual UAV1 survey resulted in $(9\pm1(1\sigma))$ minutes of useable $[X]_0$ measurements and each UAV2 survey resulted in $(8\pm1(1\sigma))$ minutes of useable $[X]_0$ measurements (see Table S1 and Table S2 for individual sampling periods). This data was prepared for flux quantification by carrying out the following steps. The $[X]_0$ timestamp from both instruments was corrected to account for lag time. 1 Hz satellite geolocation from UAV1 was interpolated to the 10 Hz $[X]_0$ frequency of the MGGA. $[X]_0$ was converted into $[X]$ using Eq. (9). $E$ was calculated with $[X]$ measurements from both instruments using Eq. (10). $[X]_b$ was derived by fitting a log-normal distribution to all recorded $[X]$ measurements from each flight survey, using the method described by Shah et al. (2019) in our previous study. This background was derived from a histogram of all useable $[X]$ measurements acquired during each flight experiment; a log-normal fit can usually be applied to the lowest $[X]$ measurements in the histogram, which represent out-of-plume sampling. The peak of the log-normal fit to these lowest $[X]$ measurements was taken to be $[X]_b$. $\rho$ was derived using average temperature, pressure and relative humidity recorded at the stationary anemometer for the duration of each flight survey, with the standard deviation in temperature, pressure and relative humidity used to derive $\sigma_\rho$.

Satellite-derived altitude was corrected to obtain the height of the air inlet above ground level, by taking into account take-off altitude and the height of the air inlet when on the ground. This step ensures that the data represent the true point of sampling. After converting longitude and latitude from degrees into meters, metric longitude and latitude were projected onto a plane perpendicular to and a plane parallel to mean wind direction, respectively. Mean wind direction was derived from the stationary anemometer for the duration of each flight survey. The coordinate projection procedure is described in further detail by Shah et al. (2019).

In order to calculate flux, flux density, $q$, (in kg s$^{-1}$ m$^{-2}$) was derived. To achieve this, each geospatially mapped $E$ measurement was combined with $WS(z)$, using Eq. (12).

(12)     $q = E \cdot WS(z)$

Geospatially mapped $q$, on a plane perpendicular to mean wind direction, for each flight survey, is plotted in Fig. 6 for UAV1 and in Fig. 7 for UAV2. Measurements of [$X$] (see Fig. 4 for a time series for each survey) were not used in the flux analysis, but are nevertheless of interest, as they show [$X$] to generally reduce with $z$, as expected, to support observations of $q$ enhancements shown in Fig. 6 and Fig. 7.

Both Fig. 6 and Fig. 7, show significant background sampling (yellow data points), extending sufficiently far away from the position of the source projected onto the sampling plane (0 m), such that the narrow turbulently advecting time-invariant plume centre across each transect (typically manifested by $q$ increase) had been passed. All of the UAV1 surveys in Fig. 6 took place from a similar distance from the source, of approximately 50 m. It is clear that during most UAV1 surveys, enhancements in $q$ were concentrated near the ground (below 10 m) and close to the position of the source, projected onto the sampling plane (0 m). However T1.3 shows considerable enhancements in $q$ above the ground (up to approximately 30 m), which was possibly due to a transient updraft. Meanwhile, the UAV2 flight surveys in Fig. 7, many of which took place approximately 100 m from the source, show large enhancements in $q$ across the flux plane, up to approximately 15 m above the ground. Enhancements of $q$ in Fig. 7 can also be seen at a much greater lateral distance from the source, projected onto the sampling plane. This is likely a consequence of many UAV2 flight surveys sampling at a greater distance from the source than UAV1 flight surveys, which gave the time-invariant plume more time to disperse. On the other hand, UAV1 flight surveys, which took place nearer to the source, show that UAV1 intersected the time-invariant plume less often. Thus, it may appear that the UAV flight track was not centred downwind of the source, when in practice erratic variations in the position of the time-invariant plume centre made it appear this way, as the time-invariant plume did not have time to disperse.

### 3.3 Flux quantification

Calculated $q$ from each flight survey was used to derive an emission flux (in units of kg s$^{-1}$) using the near-field Gaussian plume inversion (NGI) flux quantification technique (see Shah et al. (2019)). In principle, the NGI method accounts for turbulent wind variations using Gaussian statistics. The method also takes into account sampling on a slightly offset sampling plane (compared to the plane perpendicular to mean wind direction) by introducing a third dimension to the traditional two-dimensional Gaussian plume model. The NGI method uses a least-squares approach to compare measured and modelled values of $q$. Residuals in $q$ are minimised to output model parameters, which include an initial flux estimate ($F_e$).

Full details of the NGI method can be found in our previous study in Shah et al. (2019). We provide a brief overview here. The size of the time-averaged plume is assumed to increase linearly with distance from the source, by assuming $q$ to decrease according to the inverse square law with distance (an assumption which is valid over short distances). Therefore

instead of using constant crosswind and vertical dispersion terms, these terms are allowed to increase with distance from the source, with both terms being fixed at a one metre distance. The crosswind dispersion term (at 1 m) is characterised using measurements of $q$, rather than assumptions of atmospheric stability, as these assumptions are valid for time-averaged plumes characterised by dispersion, rather than turbulent advection. In addition, the centre of the time-averaged plume in the crosswind direction is derived from measurements of $q$, as the precise position of the source may be unknown. The vertical dispersion term (at 1 m) and $F_e$ can then be acquired by inverting modelled values of $q$, derived by minimising residuals, as described above.

A measurement flux uncertainty ($\sigma_F$) is calculated by combining the uncertainties in individual $E$ and $WS(z)$ values. A lower uncertainty bound ($\sigma^-$) is calculated using residuals between modelled and measured $q$ values. An upper uncertainty bound ($\sigma^+$) is calculated by incorporating $\sigma^-$ with the potential effects of negative flux bias due to under-sampling, using a random walk simulation. The simulation is repeated 180 times for each flight survey. In each simulation, a static Gaussian plume (simulating a prescribed arbitrary target flux) is sampled across three dimensions, where sampling is constrained to the spatial limits of UAV sampling and is limited to the UAV sampling duration. The NGI method is used to derive a flux from these random walk simulations. The average fractional target flux underestimation from these simulations can be incorporated into $\sigma^+$. Random walk flux underestimation occurs due to limited spatial sampling coverage (i.e. sampling gaps) and limited spatial sampling extent. This simulation step therefore gives an important indication of the systematic error due to potential under-sampling. All $F_e$, $\sigma^-$, $\sigma^+$ and $\sigma_F$ values for each flight survey are given in Table S5.

**4 Flux results and discussion**

Calculated NGI emission fluxes were compared to the known (controlled) emission fluxes, using the ratio between the NGI flux uncertainty range and $F_0$ (see Fig. 8). As this was a blind flux analysis, $F_0$ was not revealed to the analysis team prior to calculating the NGI flux uncertainty range. Fig. 8 shows that the NGI flux uncertainty range agrees well with $F_0$, for most flight surveys. Only three surveys (T2.1, T1.1 and T1.7) had a flux uncertainty range that fell short of $F_0$. Although no flux uncertainty range exceeded $F_0$, T2.3 spanned a large flux range, much of which fell above $F_0$. Flux underestimation may be explained using the plots shown in Fig. 6 and Fig. 7, which demonstrate the following: a limited sampling duration made it possible to almost entirely avoid the time-invariant emission plume, thus resulting in low flux results; similarly, some flights intersected the time-invariant emission plume multiple times resulting in flux overestimation in cases, although large NGI uncertainty ranges can conservatively account for this effect. Therefore it is clear that the $F_e$ value obtained using the NGI method must not be taken at face value and the full NGI flux uncertainty range must be considered. Furthermore, the flux ranges in Fig. 8 represent uncertainty bounds of one standard deviation; it is statistically realistic to expect some discrepancy between $F_0$ and the NGI flux uncertainty range.

The flux uncertainty ranges given in Fig. 8 are asymmetric, although the magnitude of this asymmetry was different for flight experiments conducted by the different UAVs. $\sigma^+$ was ($0.33\pm0.14(1\sigma)$) times larger than $\sigma^-$ for UAV2 but was only ($0.08\pm0.03(1\sigma)$) times larger for UAV1. This is because UAV2 sampled further from the source, on average, and on a similar sized sampling plane to UAV1. As UAV2 was further from the emission source, the time-invariant plume had a greater likelihood of extending beyond the sampling plane and being missed (beyond the horizontal edges of the sampling plane), due to spatially limited sampling extent. This potential loss of in-plume sampling may have otherwise contributed towards the overall flux, thus enhancing $\sigma^+$. Therefore $\sigma^+$ is comparatively larger than $\sigma^-$ for flights conducted by UAV2.

The suitability of our experimental sampling methodology can be assessed by quantifying $\sigma_F$ as a fraction of $F_e$, which was on average ($\pm45\pm8$)%. To assess the dominant sources of $\sigma_F$, the contribution of $WS(z)$ and $E$ uncertainty components towards it were analysed (see SI for details and results). As $\sigma_F$ is derived by combining individual components in quadrature, this analysis was conducted by assuming other uncertainties to be zero. The test showed that if wind speed was the only source of uncertainty, it would on average result in ($\pm90\pm8$)% of $\sigma_F$, therefore representing a dominant source of uncertainty. The standard deviation variability in cell temperature and cell pressure within each flight survey (see Table 5) was, on average, far smaller than maximal cell temperature changes (2.9° C) and cell pressure changes (3.5 mbar) observed during the MGGA Allan variance test. The average cell temperature and cell pressure during each flight survey was also derived (see Fig. S18 and Fig. S19) with averages given in Table 5. The values in Table 5 are not dissimilar to conditions during calibrations (plotted in Fig. S18 and Fig. S19). As there was no discernible correlation between [$X$] and cell temperature and cell pressure from the MGGA Allan variance test and considering dominance of winds contributing towards $\sigma_F$, one can assume that variation in cell temperature and cell pressure had negligible net effect on $\sigma_F$. Furthermore, the (poorly correlated) temperature trend from the MGGA Allan variance test reveals a maximum uncertainty of 20 ppb for the MGGA and 14 ppb for the pMGGA (derived from the maximum difference between average calibration cell temperature and average UAV sampling cell temperature). These uncertainty values are far smaller than the average mole fraction enhancement uncertainty (expressed as a dry mole fraction) within each flight survey of (55±47) ppb (see Table S7 for individual values), though further laboratory testing would be needed to better characterise these effects (see sect. 2.5).

It is important to recognise the magnitude of the NGI uncertainty ranges in Fig. 8, relative to $F_0$, which are due to the difficulties in inverting sparse spatial sampling to derive an emission flux, following the NGI method. These uncertainties reflect the limited sampling duration and the effects of variability in wind. While we fully acknowledge that flux uncertainty ranges in Fig. 8 are large, the true value of the NGI method with UAV sampling is to derive snap-shot rapid flux estimates at low cost, with an order-of-magnitude level precision, for subsequent flux investigation using more precise approaches. Although longer sampling periods in each flight survey may reduce the uncertainties in Fig. 8, this is practically difficult with limited UAV battery life, with little additional benefit. Tethered power or multiple UAV flights may alternatively be used, as was the case with UAV1, but wind conditions can quickly change when sampling for prolonged periods with too many lengthy intervals between flights.

Some flux results (T1.1 for example) intersected the time-invariant plume more often than others (T1.2 for example) but resulted in a lower NGI flux range. On closer inspection of the mole fraction time series given in Fig. 4, flight surveys such as T1.2 sampled higher mole fraction enhancements (and hence $q$), than T1.1. However, as the time-invariant plume may have largely been centred near to the ground, it can be more difficult to distinguish from a simple plot of the flux density UAV flight track. The comparative magnitude of mole fraction enhancements is clear, on examination of the mole fraction time series. Thus it is important to take into account both the number of plume intersections and the magnitude of $q$ during each plume intersection, when assessing NGI flux results.

In order to assess whether multiple flight surveys could be used effectively to capture the known controlled emission flux, within uncertainty, the upper and lower NGI uncertainty bounds were averaged for all surveys (see penultimate row of Fig. 8). The average lower NGI flux uncertainty bound as a fraction of $F_0$ ($\overline{F_-}$) was 0.2±0.1(1$\sigma$) and the average upper NGI flux uncertainty bound as a fraction of $F_0$ ($\overline{F_+}$) was 2±1(1$\sigma$), for all surveys. Thus $F_0$ (i.e. 1 in Fig. 8) falls comfortably within the average NGI flux uncertainty range, over 22 independent flight surveys. $\overline{F_-}$ and $\overline{F_+}$ were also calculated for surveys conducted by UAV1 and UAV2, separately. These separate $\overline{F_-}$ and $\overline{F_+}$ values for each UAV also comfortably overlap with

the $\overline{F_-}$ and $\overline{F_+}$ values for all surveys combined. This suggests that the sampling strategies employed by both UAVs were capable of deriving the known emission flux, with a similar degree of both lower and upper uncertainty. The percentage standard error in $\overline{F_-}$ and $\overline{F_+}$, over all 22 flight surveys, was 12% and 9%, respectively. The large standard errors in $\overline{F_-}$ and $\overline{F_+}$ may be reduced with more surveys, in order to better constrain the NGI flux uncertainty range. However more precise flux estimates can be obtained using other approaches such as tracer dispersion methods. Although we recognise that the $\overline{F_-}$ and $\overline{F_+}$ uncertainty averages are large, we emphasise that our methodology has been adapted for rapid flux analysis, rather than precise flux estimates for inventory publication.

The ability of the NGI method to calculate a target emission flux was further assessed by calculating the central flux estimate as a fraction of $F_0$ ($F_c$) for each flight survey, using Eq. (13). $F_c$ is distinct from $F_e$ (as a fraction of $F_0$), in that $F_c$ finds the centre of an asymmetric flux uncertainty, whereas $F_e$ is an initial flux estimate calculated using the NGI method, which does not take into account the potential effects of under-sampling, which may result in a potential negative flux bias.

(13)    $$F_c = \frac{F_e + \left(\frac{\sigma_+ - \sigma_-}{2}\right)}{F_0}$$

The mean of $F_c$ ($\overline{F_c}$) and the mean standard error in $\overline{F_c}$ for the 22 surveys (see bottom row of Fig. 8) treats each survey as an independent quantification of the flux, with no weighting for sampling time (as flight times were broadly similar). This clearly demonstrates the improvement in flux accuracy (for a constant source) that can be obtained with greater sampling time or repeated flights, as expected. $\overline{F_c}$ was also calculated for surveys conducted by UAV1 and UAV2 separately: these separate $\overline{F_c}$ values both overlap with the combined $\overline{F_c}$ value for all flight surveys (within one standard deviation); there is no discernible difference in the NGI flux results obtained by either UAV. This suggests that both UAV sampling strategies were equally capable of delivering the same emission flux estimate, by taking the average of multiple flight surveys.

The overlap of the standard deviation in $\overline{F_c}$ (shown in Fig. 8) with the known emission flux (i.e. 1 in Fig. 8) also suggests that there was no apparent flux bias (within uncertainty) in this study. This indicates that we have successfully overcome the causes of positive biases reported in our previous study (Shah et al., 2019). Shah et al. (2019) sampled downwind of a controlled emission source and actively pursued the time-invariant emission plume (projected onto the sampling plane) using mid-flight knowledge of its position, inferred by releasing smoke grenades during flight surveys. However in this current work, manual sampling was avoided by either flying UAV1 using pre-programmed waypoints or by flying UAV2 using lateral transects in course-lock. Both of the approaches presented here successfully avoided biased sampling.

To conclude, UAV sampling can be used to practically derive unbiased snap-shot emission fluxes with the NGI method, with an order-of-magnitude precision, by sampling on a plane perpendicular to wind direction from at least approximately 50 m away from the source. Although typical flux uncertainties were high, NGI UAV fluxes serve as an important tool for snap-shot source identification and flux quantification. Our UAV methodology fills an important gap between cheap leak detection techniques (such as infrared cameras), which do not provide fluxes, and reliable flux quantification techniques (such as the tracer dispersion method), which require expensive instrumentation and may be more difficult to organise. For example, tracer methods can be problematic in cases where site access for tracer release is impossible or in cases where the plume may be lofted. The UAV methodology we describe is highly suitable for regulatory leak detection and source isolation, with the added capability to gauge the severity of flux leaks, for subsequent investigation using other approaches. We anticipate a combination of UAV sampling with a tracer release, where both a target gas (in this case methane) and a proxy tracer can be measured simultaneously downwind, taking advantage of vertical sampling enabled by UAVs, as a powerful future toolkit for precise facility-scale flux quantification.

**5 Conclusions**

Two UAVs were used to test the near-field Gaussian plume inversion technique for flux quantification. One UAV was connected to the MGGA on the ground using a tether, while the other carried a new ABB pMGGA prototype instrument on-board. Both instruments measured atmospheric methane mole fraction, which was calibrated and corrected for the influence of water vapour, following laboratory testing under ambient conditions, assuming the effects of cell temperature and cell pressure to be small.

The flux approach was tested for 22 UAV flight surveys, by deriving fluxes from a controlled release of methane gas. This yielded successful results, with 19 out of 22 fluxes falling within the UAV-derived flux uncertainty range. This demonstrates that the near-field Gaussian plume inversion methodology used here could be used to derive emission fluxes from UAV sampling of plumes from facility-scale (point) sources, where such sources are relatively invariant over the period of such UAV sampling. The lower flux uncertainty bound was, on average, $17\%\pm10(1\sigma)\%$ of the controlled emission flux and the upper flux uncertainty bound was, on average, $227\%\pm98(1\sigma)\%$ of the controlled emission flux. Thus the known emission flux was comfortably encapsulated by the UAV flux results, within uncertainty.

A key advantage of the methodology used here is the ability to sample downwind of sources to obtain off-site mole fraction measurements. Such sampling allows for independent and portable studies of methane emissions without the need for heavy infrastructure, special permissions, runway access or prior notification. We conclude that the near-field Gaussian plume inversion flux quantification method can be used confidently in future with UAV sampling to derive snap-shot methane emission fluxes from relatively constant facility-scale sources such as oil and gas extraction infrastructure, livestock agriculture and landfill sites. An exciting future application may be the incorporation of UAV sampling within a tracer release methodology, where simultaneous measurement of a target gas and a proxy tracer can take advantage of vertical sampling enabled by UAVs. This avoids the limitation of current mobile vehicle sampling which cannot sample lofted plumes. Together, this may represent a powerful future toolkit for precise and efficient flux quantification.

**Author contribution**

AS, JRP, HR, JBL, PIW and GA carried out the field experiments. All authors designed the field experiments. AS and JRP carried out and designed the laboratory experiments. AS, JRP and JBL developed the sensor characterisation procedures. JBL provided access to the prototype sensor. AS wrote the manuscript. GA edited the manuscript. All authors contributed towards the manuscript.

**Acknowledgments**

Adil Shah's PhD studentship is funded by the Natural Environment Research Council (NERC), grant reference NE/L002469/1, and is supported through a CASE partnership with the Environment Agency. NERC also provided contributions, in kind, through EQUIPT4RISK, grant reference NE/R01809X/1. We thank ABB for loaning us the pMGGA for this work. We also thank the farmer for allowing us to operate on his land during sampling and Cuadrilla Resources Ltd. for facilitating access.

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

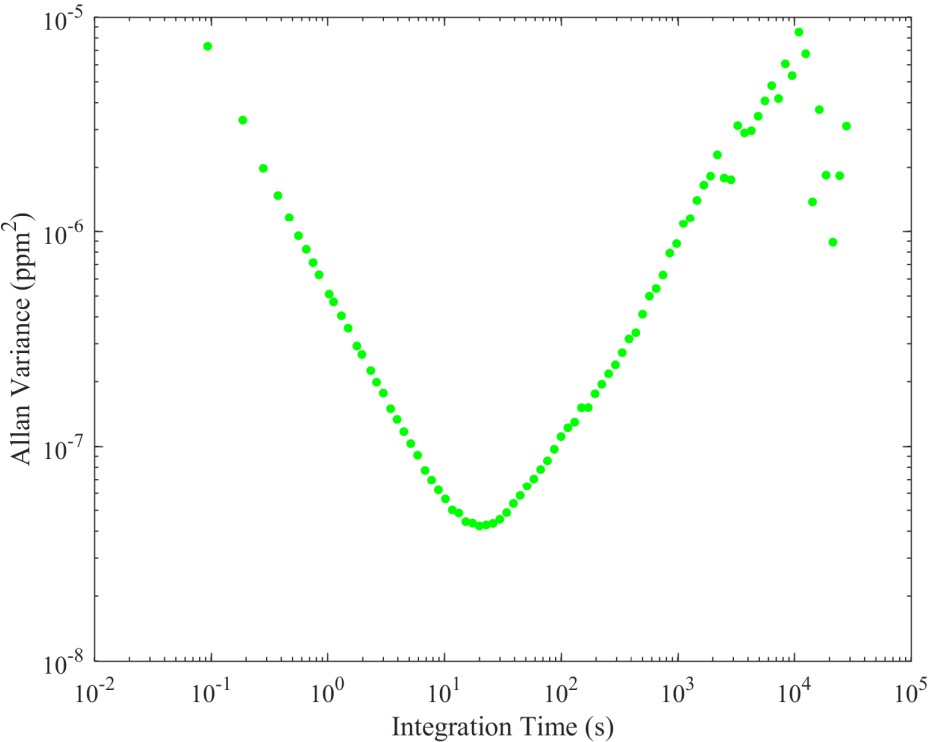

**Figure 1. Allan variance for the MGGA plotted against integration time on logarithmic axes.**

715

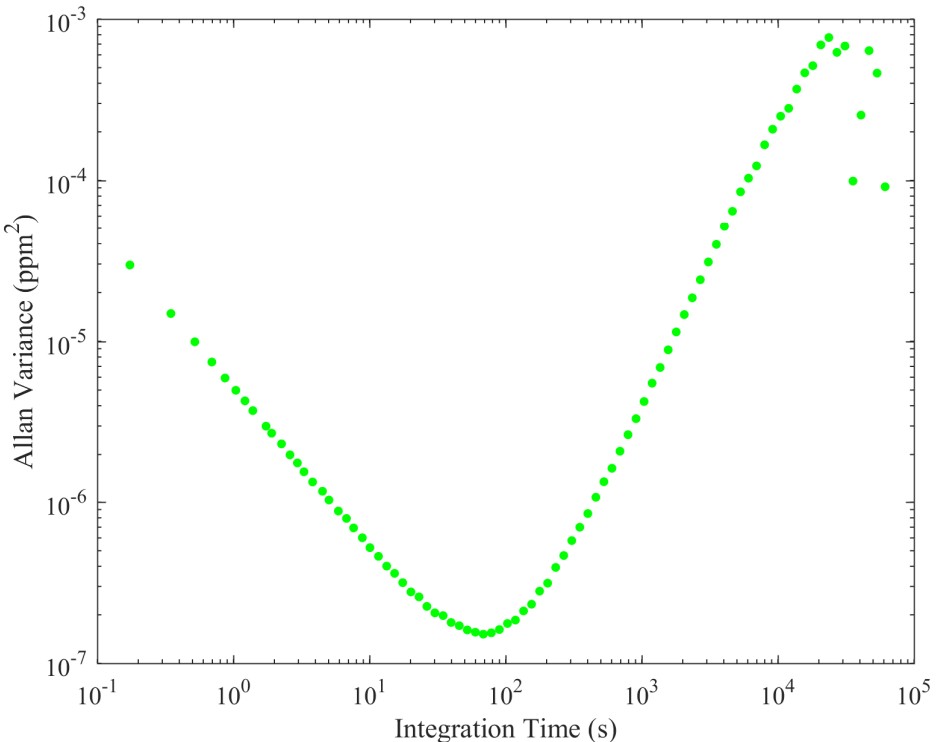

**Figure 2. Allan variance for the pMGGA plotted against integration time on logarithmic axes.**

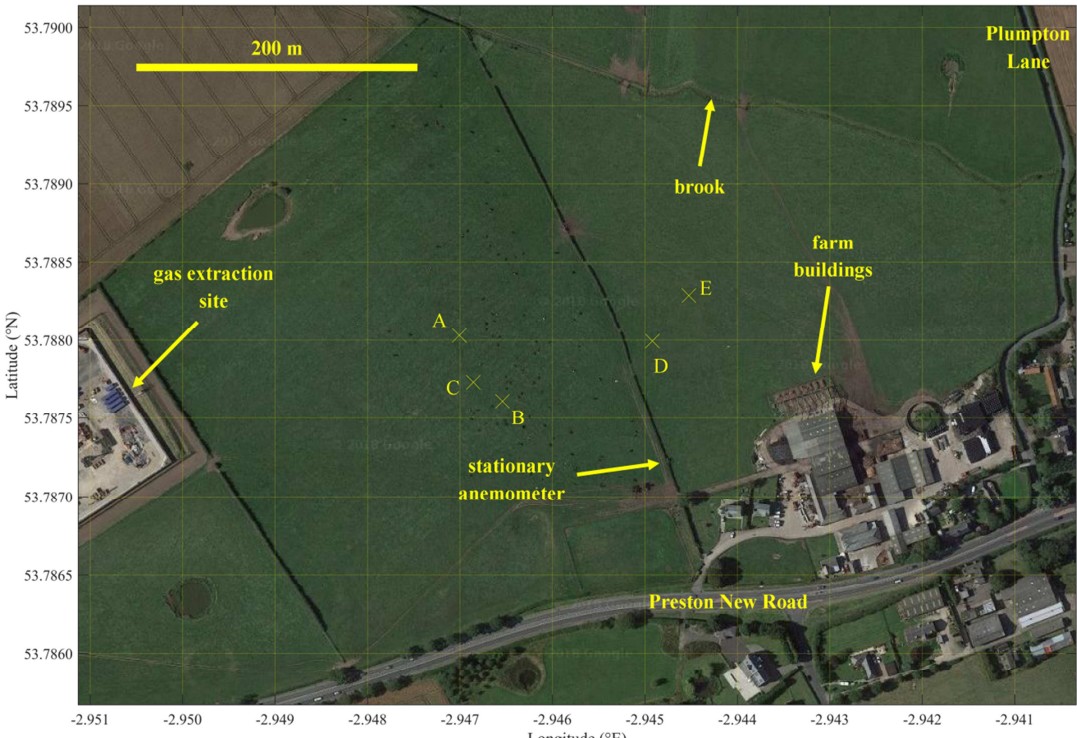

720     **Figure 3. The two fields used for UAV sampling. The map extends 0.71 km horizontally and 0.50 km vertically. The controlled release points are marked by labelled crosses (see Table S3 for details). The background image is taken from Google Maps (imagery (2017): DigitalGlobe, GetMapping plc, Infoterra Ltd & Bluesky, The GeoInformation Group).**

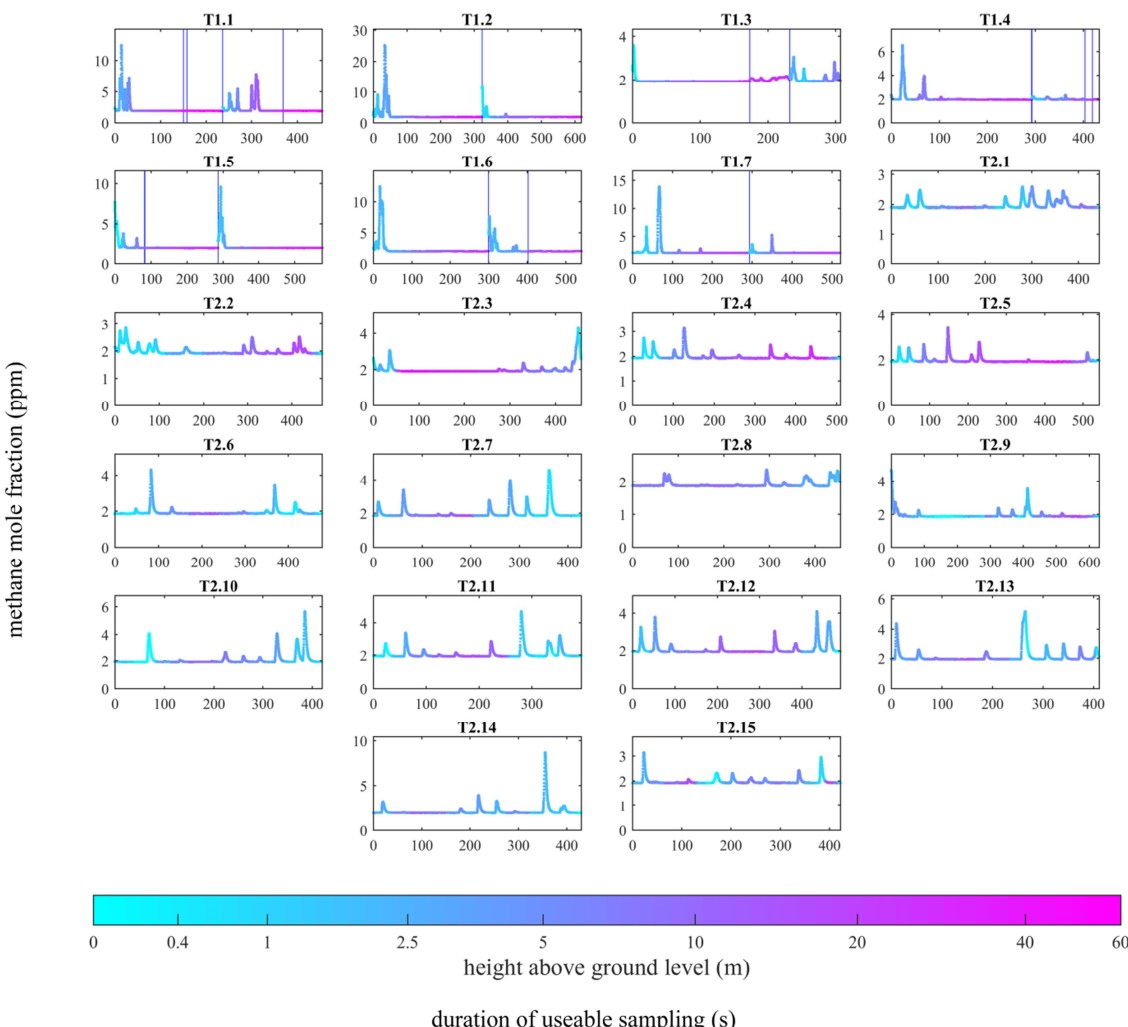

725

**Figure 4. [*X*] measurements acquired by the MGGA and the pMGGA, as a function of sampling duration, for each flight survey, with sampling height above ground level also plotted (coloured dots). A logarithmic colour legend has been used. Vertical blue lines indicate an interruption in continuous sampling.**

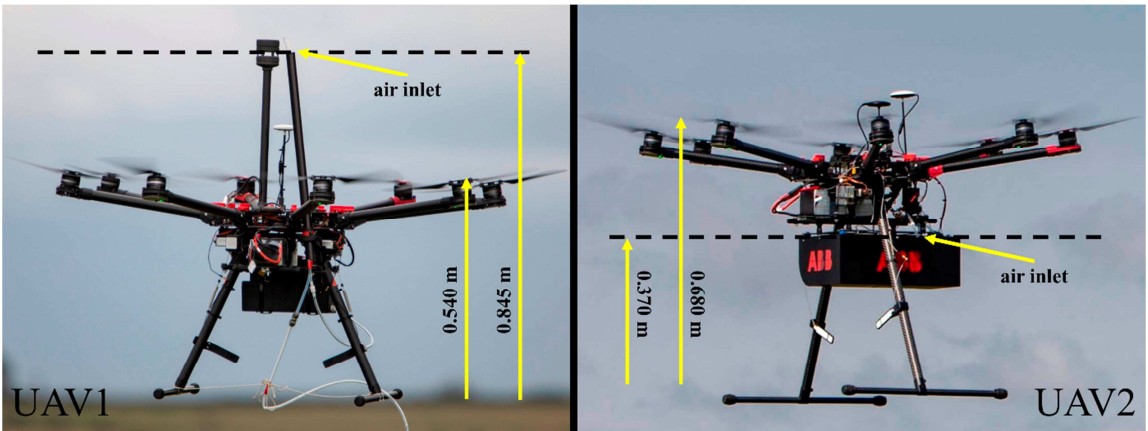

**Figure 5. A photograph of UAV1 and UAV2, indicating the position of the air inlet relative to the base of the UAV.**

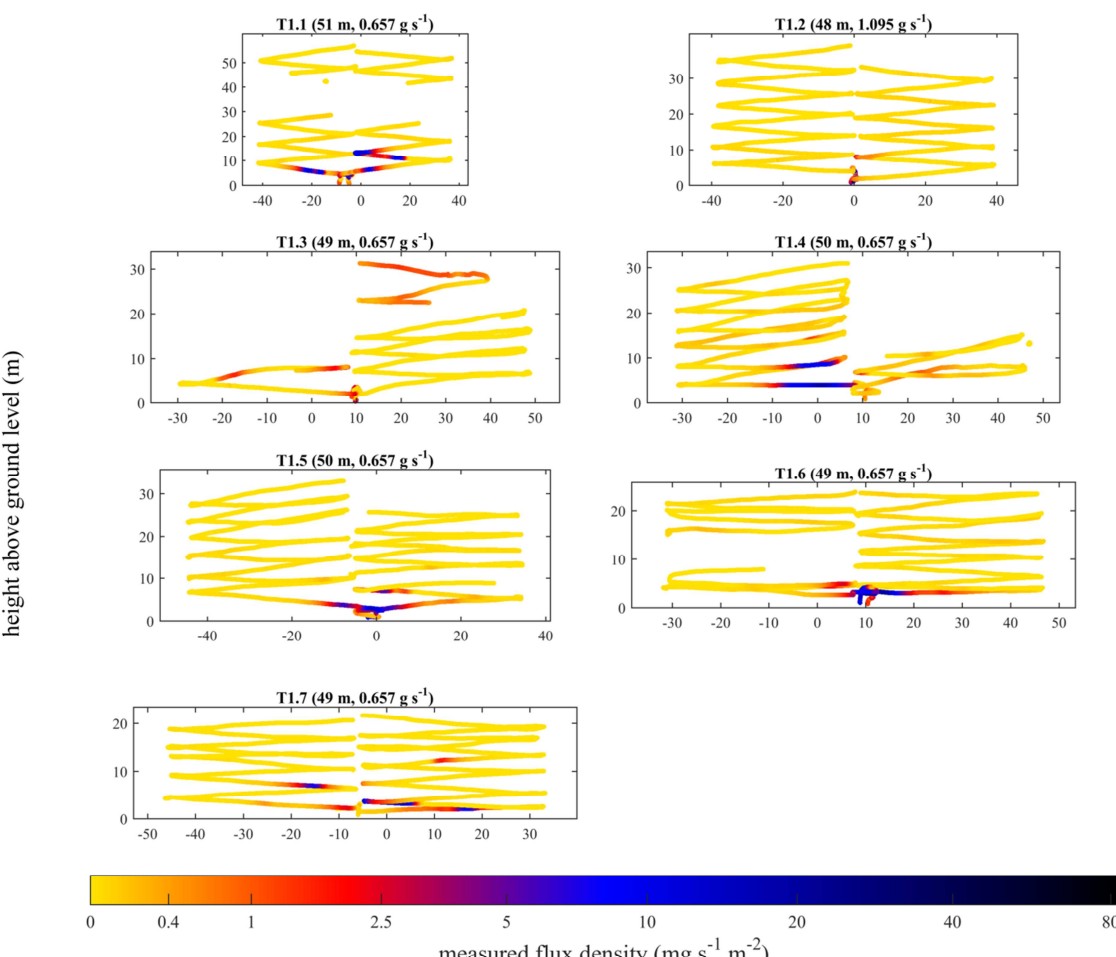

distance along plane perpendicular to mean wind direction (m)

**Figure 6. UAV1 flight tracks (coloured dots), with the colour corresponding to $q$. Periods in which the tubing inlet kinked have been removed. A logarithmic colour legend has been used. The position of the source projected on the plane perpendicular to mean wind direction has been set to a reference of 0 m. The controlled emission flux and the parallel distance of the sampling plane from the source (weighted to the position of $q$ enhancements) are given in brackets.**

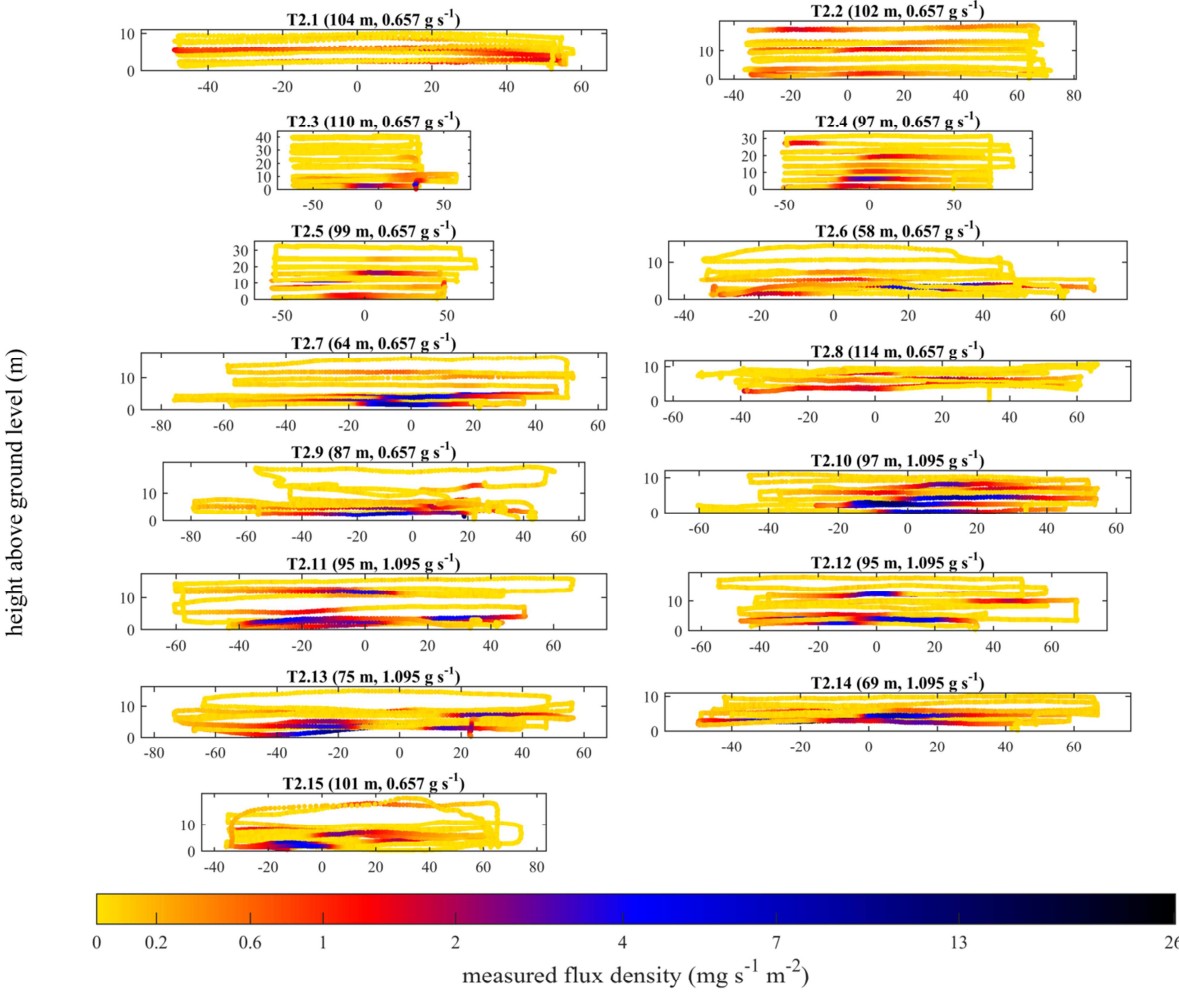

height above ground level (m)

distance along plane perpendicular to mean wind direction (m)

**Figure 7. UAV2 flight tracks (coloured dots), with the colour corresponding to $q$. The position of the source projected on the plane perpendicular to mean wind direction has been set to a reference of 0 m. The controlled emission flux and the parallel distance of the sampling plane from the source (weighted to the position of $q$ enhancements) are given in brackets.**

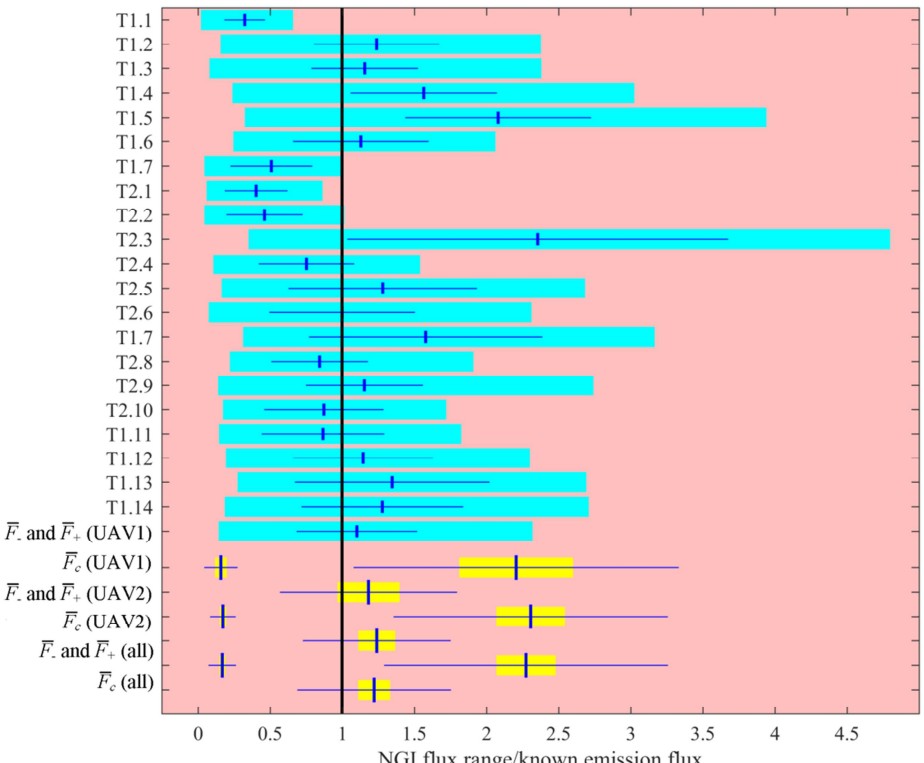

**Figure 8. NGI flux uncertainty range (thick cyan bars), for each method testing flight survey, as a fraction of $F_0$. The $\sigma_F$ uncertainty range (horizontal blue lines) is given on either side of $F_e$ (vertical blue lines). $\overline{F_c}$ and $\overline{F_-}$ and $\overline{F_+}$ averages (vertical blue lines) are plotted for UAV1, UAV2 and for all flight surveys. Standard deviation uncertainty ranges (horizontal blue lines) and standard error uncertainty ranges (thick yellow bars) are given on either side of $\overline{F_c}$, $\overline{F_-}$ and $\overline{F_+}$ values.**

| | MGGA | pMGGA |
|---|---|---|
| **Mass** | 4.8 kg | 3.4 kg |
| **Length** | 0.35 m | 0.33 m |
| **Width** | 0.30 m | 0.20 m |
| **Depth** | 0.15 m | 0.13 m |
| **Power consumption** | 35 W | 32 W |
| **Operating DC voltage** | 10 V – 30 V | 10 V – 28 V |
| **Cell pressure** | atmospheric | pressure controlled to 0.61 bar |
| **E-folding time** | $(1.6\pm0.2)$ s | $(3.0\pm0.1)$ s |
| **Volumetric flow rate** | $(27.90\pm0.05)$ cm$^3$ s$^{-1}$ | $(5.08\pm0.02)$ cm$^3$ s$^{-1}$ |
| **Maximum sampling frequency** | 10 Hz | 5 Hz |
| **$\sigma_n$** | $\pm2.71$ ppb | $\pm5.44$ ppb |
| **1 Hz Allan deviation** | $\pm0.71$ ppb | $\pm2.2$ ppb |
| **0.1 Hz Allan deviation** | $\pm0.24$ ppb | $\pm0.72$ ppb |
| **Optimum integration time** | $(20\pm3)$ s | $(70\pm10)$ s |

**Table 1: General properties of the MGGA and the pMGGA.**

755

| | MGGA | pMGGA |
|---|---|---|
| *a* | -0.000312 $\text{mol}_{water}$ $\text{mol}^{-1}$ | +0.000195 $\text{mol}_{water}$ $\text{mol}^{-1}$ |
| *b* | -0.000193 $\text{mol}_{water}$ $\text{mol}^{-1}$ $\text{ppm}^{-1}$ | -0.0000257 $\text{mol}_{water}$ $\text{mol}^{-1}$ $\text{ppm}^{-1}$ |
| *α* | -1.556 $\text{mol}$ $\text{mol}_{water}^{-1}$ | -1.640 $\text{mol}$ $\text{mol}_{water}^{-1}$ |
| *β* | -12.25 $\text{mol}^2$ $\text{mol}_{water}^{-2}$ | -1.208 $\text{mol}^2$ $\text{mol}_{water}^{-2}$ |
| *σ$_v$* | 0.0004253 | 0.0002613 |

**Table 2: Water correction coefficients for the MGGA and pMGGA, required to obtain *v* using Eq. (4) and *σ$_v$*.**

|  | MGGA | pMGGA |
|---|---|---|
| $G \pm \sigma_G$ | 0.9970±0.00023 | 0.9869±0.00028 |
| $C \pm \sigma_C$ | (+0.0132±0.0020) ppm | (-0.0019±0.0015) ppm |

**Table 3: Calibration coefficients for the MGGA and pMGGA.**

|  | UAV1 | UAV2 |
|---|---|---|
| **Flights per survey** | 2 | 1 |
| **Distance of sampling plane from source** | 47 m – 50 m | 64 m – 114 m |
| **Take-off and landing** | Manual | Manual |
| **Flight control** | Waypoints | Manual (course lock) |
| **Average velocity across the sampling plane** | $(1.5\pm0.1)$ m s$^{-1}$ | $(2.8\pm0.6)$ m s$^{-1}$ |
| **Payload** | PFA tubing and inlet, wind sensor | pMGGA |
| **Height of plane of propellers** | 0.540 m | 0.680 m |
| **Height of air inlet** | 0.845 m | 0.370 m |

760    **Table 4: A comparison between UAV1 and UAV2.**

|  | UAV1 | UAV2 |
|---|---|---|
| **Instrument** | MGGA | pMGGA |
| **Average cell temperature standard deviation within flight surveys** | (±0.16±0.09)° C | (±0.28±0.28)° C |
| **Average cell pressure standard deviation within flight surveys** | (±1.15±0.86) mbar | (±0.40±0.01) mbar |
| **Average cell temperature mean across flight surveys** | (25±2)° C | (22±4)° C |
| **Average cell pressure mean across flight surveys** | (1025.6±5.4) mbar | (614.4±0.1) mbar |

**Table 5: Average cell temperature and cell pressure standard deviation variability within each UAV flight survey, recorded by the MGGA and the pMGGA. The average cell temperature and cell pressure mean is also given.**