# Peer review of "Testing the near-field Gaussian plume inversion flux quantification technique using unmanned aerial vehicle sampling"

_Atmospheric Measurement Techniques, 2019_

## Referee Comment (RC1) · Anonymous Referee #1 · 6 Nov 2019

Shah et al. employ two different UAV platforms to quantify known sources of CH4 during a series of release experiments. The deployment of a lighter prototype Micro-portable Greenhouse Gas Analyzer (pMGGA) is new and may be interesting to other potential users as well. The authors have done a reasonable job to characterise the sensor in the laboratory; however, testing of the sensor in a harsh environment with varying temperature and pressure is missing. The near-field Gaussian plume inversion methodology presented in previous work was applied to the release experiments, and the estimated fluxes and associated uncertainties were compared with the known sources. The paper is well structured and well written, and can be published at AMT after taking into account the following comments.

[Figure]

General comments:

1. A weak point of the calibration in Section 3.2 is that the temperature and pressure dependence of CH4 measurements by both the MGGA and the pMGGA is not characterised, which may be potentially much larger than the gain factor uncertainty and the offset uncertainty. In the case that the field characterisation was not performed, why not characterize it in the laboratory?

2. What's the reason behind the exponential decay of H2O with CH4 mole fraction? Is it due to line interferences? It is difficult for readers to judge when the wavelengths of H2O, CH4 are not given. Is there an interference between CO2 and H2O as well? Notice that the exponential fits in Figures S3&S4 are based on very limited data points. What's the air matrix of the 100 ppm CH4 cylinder? Could the dependence of H2O measurements be caused by other species?

3. Regarding the uncertainties of the estimated fluxes $\sigma$F, what are the fractional contributions due to individual components? This information may help reduce the uncertainties in future measurements.

Detailed comments:

L92-93: The flow rate needs to be given when the e-folding time is discussed. Alternatively, the e-folding volume can be provided.

L97-98: The unit should be ppb instead of ppm.

L159-162: should make it clear that +0.27% and +1.8% are the differences between with and without the water vapour corrections, instead of an increase of measurement accuracy.

L207: Equation 10 should use the molar density of dry air since CH4 is given in dry mole fraction.

L234-239: what was the nominal flow rate? Was the flow rate recorded? What fraction

of measurements on average were omitted from each flight?

L239-240: what was the flow rate through the pMGGA?

L326-330: Comparing T1.1 with T1.2 in Figure 3, I expect that larger emissions would be quantified for T1.1 and with larger uncertainties, however, the results showed the opposite. Why is that? Where are the centers of the plumes found?

L335-341: It looks that the crosswind distance is not sufficient to cover the plume, especially for UAV1. Why were the transects of UAV1 not centered? The current sampling tends to miss the center of the plume.

---

## Referee Comment (RC2) · Anonymous Referee #2 · 14 Nov 2019

Review

Testing the near-field Gaussian plume inversion flux quantification technique using unmanned aerial vehicle sampling

Adil Shah et al.

The authors use a near-field Gaussian plume inversion (NGI) technique to determine the strength of methane emission sources. In that context, they describe, characterize and deploy two commercial near-IR off-axis integrated cavity output spectroscopy (ICOS) spectrometers from ABB/Los Gatos Research. One instrument is ground-based and samples through a tube from the UAV, while the other, more lightweight instrument is deployed onboard the UAV. The authors rightly claim that more reliable techniques are needed to determine emissions from diffuse methane sources. The paper has two main technical sections: chapter 2 "Methane instrumentation and calibration" and chapter 3 "Method Testing". After struggling hard with chapter 2, I decided not to continue the review, and I suggest that the paper must be fundamentally rewritten, with respect to both language and content, before publication in AMT.

Chapter 2 has four subsections:

2.1 Instrument inter-comparison (20 lines) Jonas, please check # lines

2.2 Water vapour correction (61 lines)

2.3 Calibration (28 lines)

2.4 Methane enhancement and uncertainty (20 lines)

Section 2.1 d contains very little actual intercomparison, and thus the title is misleading The inter-comparison consists mainly of one table containing the manufacturer's specifications and the key numbers of one Allan Variance plot (based on laboratory measurements), which is given in S1. In the context of this paper, which attempts to use the instruments under harsh field conditions, much work could and should have been done to provide useful, real-world characterization under field conditions, beyond what the manufacturer reports. This may include (but should not be limited to) performance characteristics and comparison (see section title) of long-term stability, stability of calibration factors, temperature and pressure dependence, sensitivity to vibration (UAV), linearity, selectivity, (measured) response time, concentration range, etc.

Section 2.2 is the core part of the instrumental section, which is reflected by its length. It covers an in-depth description of water vapor correction but sadly contains a number of fundamental flaws. Some examples are below, but this list is not exhaustive:

- The largest correction to obtain dry $CH_4$ values ($[X]^{dry}$), needed for flux calculations, is water vapor. While the authors do many tests and calculations (see below) to get this value, they never show how accurate the $H_2O$ measurements of the instruments are. They likely have the information

(based on the measurements with a dew point generator), but the corresponding performance is never shown. For such a detailed water vapor study, it would be key to deduce this effect first and then discuss (and correct) the remaining influencing parameters in a second step.

- Figure S3 shows the effect of $CH_4$ on apparent water, due to cross sensitivity of the analyzers. Equation 1 has no physical meaning or motivation, but it produces an exponential curve, which fits the data points nicely. If there is no clear motivation, such a weak set of data may as well be described by a linear interpolation (see S3). Furthermore, applying the same parameter ($\omega$) for the other analyzer, which shows clearly different absolute and relative behavior (S4), is not justified.

- Somewhat similar is the issue in S5 and S7. It is not surprising to find a (nearly linear) dependence between $\nu$ and water vapor. It is also not surprising that the residuals become a bit smaller if using a second order polynomial. However, since there is no information on the long-term stability of this system (e.g. repeatability and reproducibility), these fit characteristics are of very little use (a larger order polynomial would give even smaller residuals).

- The authors call $\nu$ the water correction factor. This, again, is misleading because it contains normalization to dry conditions, which is a standard procedure, and corrections for non-perfect spectroscopy (which is specific to their analyzers).

- Based on these badly justified fits and corrections, the authors deduce a water uncertainty factor ($\sigma_\nu$). However, $\sigma_\nu$ is the standard deviation of the mean of the residuals in S6/7. This parameter carries information about the quality of the fit of the polynomial, but no information about the quality of the water correction for a specific measurement at a given $CH_4$ and $H_2O$ mole fraction.

Sections 2.3 and 2.4 describe the calibration procedure and derive measurement uncertainty.

- Calibrated enhancement in methane mass density (E) is calculated based on parameters that are obtained in the above experimental and fit procedures. The procedures themselves are correct (but see some remarks above and below). The authors then state that their approach is especially useful because it can be used to determine measurement uncertainty (equation 11). However, this uncertainty calculation is based on laboratory measurements and badly motivated fit procedures. It does not contain non-linearity nor any changes (drifts) in time, e.g. of G and $\nu$. Furthermore, it does not include any field (in)stability considerations, nor the fact that one parameter is just assumed to be the same in MGGA and pMGGA. A realistic uncertainty budget would be highly welcome, but the approach chosen here is useless.

Some less fundamental comments are below, illustrating that the whole paper should be strongly revised:

Abstract

Page 1 – Line 12

Many parameters contribute to the uncertainty. Are they large or small, and is the contribution due to the "maybe" poorly quantified sources important or neglectable?

Page 1 – Line 14

This is not a NEW near-field Gaussian plume inversion (NGI) technique – it has been published before by the same authors.

Page 1 – Line 20

Modified with respect to what?

Page 1 - Line 23

Simplify sentence, e.g. the uncertainty was between … and …

Page 1 – Line 25

The term "range" implies that it is the range between 17 and 218 %, but this is not meant here.

Page 1 – Line 27

"flux approaches": check language

Page 1 – Line 27

Replace "may perform well" by "may be a valuable alternative"

Page 1 – Line 28

"applied to UAV sampling" should read "combined with UAV sampling"

Introduction

Page 2 – Line 53

precision should read accuracy or precision and accuracy

Page 2 – Line 54

have not yet

Page 2 – Line 60

This suggests that controlled release is the only acceptable method. Make statement more general, e.g. the method was not validated, and its uncertainty was not quantified (e.g. with controlled release experiments).

Page 2 – Line 62

Make it clear that "Shah et al" is not any other researchers but the same group, e.g. our previous study...

Page 2 – Line 65

Language: the sampling does not develop the technique.

e.g.: ... sampling ... used in combination with ...

Methane Instrumentation and Calibration

Page 3 – Line 97

$\sigma_{AV}$ is a very unusual parameter. Contains very little information because it depends on averaging time. "Allan deviation uncertainty factor" is confusing.

Page 3 – Line 108

"These three effects have a net effect of decreasing $[X]_0$ in both instruments". You probably mean "decreasing $[X]$". Furthermore, this is only useful information if you state before which effects have a positive/negative impact. Whether the sum is a net negative effect depends on the conditions (usually dilution dominates, so the statement is not wrong).

Page 3 – Line 110

If the pressure broadening coefficient should account for effects two AND three, then state so explicitly.

Page 4 – Line 122

Has a calibration for water vapor been performed, and what are the results?

Page 4 – Line 132

"more points" means: 1.9, 2.1, 5.0 and ca. 104 ppm. It does not make sense to derive parameters of a (complicated) exponential function if there is no physical concept that supports this function.

This $H_2O$ baseline is very large; your data in S3 suggests that 100 ppm (dry) $CH_4$ results in ca. -5500 apparent ppm $H_2O$. Is this analyzer malfunctioning and should be returned to the manufacturer, or is it within the specifications for water vapor measurements? Using the MGGA $H_2O$ baseline function "as-is" for another analyzer does not make sense, especially since the experimental values for the pMGGA (S4) are significantly different from those of the MGGA, indicating that the two functions may well not be the same. Alternatively, one may just acknowledge this fact and do a corresponding uncertainty budget. In the context of the large uncertainty of flux estimation with NGI, this may be appropriate. However, it would fundamentally change the arguments and line of thought in this paper.

Page 4 – Line 135

"Having established a well characterised water baseline..."

This baseline may be well characterized. However, it corresponds to a very large correction, which cannot easily be redone in the field. Therefore, it is necessary to measure the longterm stability of the baseline and instrument-to-instrument variability. Otherwise the "well characterized" is not helpful.

Page 4 – Line 136

What was the mole fraction of $CH_4$ in this cylinder?

The $H_2O$ baseline has a huge dependence on $CH_4$. Nevertheless, this experiment was only done at one $CH_4$ concentration only. This may not be representative and add uncertainty to the retrieval of the "real-world" measurements.

Page 4 – Line 150

The difference between wet and dry corresponds to humidity, thus typically 1-3 %. This is not "almost equal". If it has little impact on the result, then this is because the retrieved apparent $H_2O$ does not very strongly depend on $CH_4$ under the conditions used to test the simplification.

Page 5 – Line 156-157

The uncertainty of water measurements cannot be derived using "the water correction residual (R) from Eq. (2)". The calculated value is not the uncertainty of water measurements, nor the uncertainty after water correction. It is the standard deviation of the mean of the residuals. This parameter carries information about the quality of the fit of the polynomial, but no information about the quality of the water correction for a specific measurement at a given CH4 and HO mole fraction.

Page 5 – Line 160

"For example our correction was used to increase 160 the MGGA measurement accuracy of $[X]_0$ (at 2 ppm) by +0.27%, at a humidity of 0.001 mol water $mol^{-1}$, and by +1.8%, at a humidity of

0.01 mol water $mol^{-1}$". This is a meaningless statement. It does not consider that any correct measurement includes the correction of the dilution effect. You compare apples with rotten pears.

Calibration

Page 5 – General Comments on Section/Calibration
Useful:

The alternating measurement of two standards. This gives the repeatability of measurements under laboratory conditions. A simple average and standard deviation would characterize this. Alternatively (x - average x) vs. N (in bins) as a figure would illustrate that the values show a normal distribution.

Not useful:

The concept of uncertainty in C and G because it neglects the fact that the linearity in response is most frequently the key factor determining the uncertainty of the measurement.

Needed:

Determination of linearity.

Page 5 – Line 168-169

This does not make sense. Water vapor in reference gas cylinders should be very low if filled dry (which I assume was done here). Even if filled wet, then 60 bar/20°C should be significantly below 500 ppm. Thus, $H_2O$ from a pressurized bottle (if treated correctly) is so low that it has very little impact on the measurements. An additional trap (as used here) may well lead to water contamination if one is not very careful.

Page 6 – Line 195-196

C deviates by about 12 ppb from 0. Whether that is good or bad (and under what conditions) depends on the application. However, to say that this is "almost equal to 0" for an analyzer with an Allan Minimum below 1 ppb is bizarre.

Page 6 – Line 195

ν is ca. 0.98 at 1 % H2O. This is not "almost equal to 1". Furthermore, ν in S6 contains all water-related corrections, including normalization to dry conditions. Stating that this is "almost equal to 1" does not make sense, because the correction to dry conditions should be done in any case.

Additional remarks

"A lightweight wind sensor (FT205EV, FT Technologies Limited) was mounted on-board UAV1, on a carbon fibre pole 305 mm above the plane of the propellers. It recorded wind speed and direction at 4 Hz. This data was used to model the change in wind speed with height above ground level (z)."

- Was this sensor validated?
- Are the measurements independent of the propellers? How was this validated?
- How reliable were the direction and speed when the drone was moving? How was this determined?

One analyzer was using an inlet just underneath the drone. The corresponding sample is representative of the air above the drone because of downwash. Was this considered? How?

Flux of one drone --> the other just sampled further away.

The Allan Variance plot should be in main section of the paper. Its v-shape is relevant for interpreting the measurements during flight (i.e. 10 – 30 minutes, typically).

---

## Author Comment (AC1) · 6 Dec 2019

We thank the reviewer for taking the time to review our manuscript and for some very useful and constructive comments and suggestions. We have responded (red text) to each comment (black italic text) in turn below. We have used these comments and suggestions to update and improve a revised version of the manuscript.

*Shah et al. employ two different UAV platforms to quantify known sources of CH4 during a series of release experiments. The deployment of a lighter prototype Microportable Greenhouse Gas Analyzer (pMGGA) is new and may be interesting to other potential users as well. The authors have done a reasonable job to characterise the sensor in the laboratory; however, testing of the sensor in a harsh environment with varying temperature and pressure is missing. The near-field Gaussian plume inversion methodology presented in previous work was applied to the release experiments, and the estimated fluxes and associated uncertainties were compared with the known sources. The paper is well structured and well written, and can be published at AMT after taking into account the following comments.*

We agree that the importance of temperature and pressure variability were not well described. We hope we have now strengthened the manuscript by addressing this concern. We have added a new table (Table 5), which shows that variability in cell temperature for the MGGA and pMGGA was no more than 1.3° C and 2.7° C, respectively, within each UAV flight, compared to a variability of 2.9° C noted during our 19-hour MGGA Allan variance characterisation. During the MGGA Allan variance test, there was no discernible systematic effect of either cell temperature or cell pressure on measured methane mole fraction (see section S1 of the supplement, newly added, which now describes this). With regard to pressure specifically, all measurements were made at ambient atmospheric surface pressure, which changes only very slightly up to 50 m in height and between flight surveys. Furthermore, the pMGGA was closely controlled to a fixed cell pressure. Regarding temperature and pressure conditions during sampling, we would argue that they were not "harsh" (see Table 5) and that we should have made this clearer in the original manuscript. During UAV sampling, recorded cell temperature was on average 23° C across all UAV flights compared to calibration temperatures for the MGGA and pMGGA of (31.4±0.7)° C and (24.6±0.1)° C, respectively. We thank the reviewer for raising this point. On a wider point, a truly harsh environment, such as volcanic caldera sampling, would (we agree) likely require further characterisation in extreme temperatures, but this is out of scope of this work.

*General comments:*

*1. A weak point of the calibration in Section 3.2 is that the temperature and pressure dependence of CH4 measurements by both the MGGA and the pMGGA is not characterised, which may be potentially much larger than the gain factor uncertainty and the offset uncertainty. In the case that the field characterisation was not performed, why not characterize it in the laboratory?*

This is a very useful and valid point. To test the effect of changes in cell temperature, we measured a test gain factor of 0.9979 (at 12.7° C lower than the main calibration) and for cell pressure, we measured a test gain factor of 0.9967 (at 37.2 mbar lower than the main calibration), in the MGGA (noting, as above, that cell pressure is fixed and controlled for the pMGGA). Neither test revealed a discernible change in MGGA gain factor within uncertainty (this point is now made in the final paragraph of section 2.3). To put this into context, if sampling 12.7° C above the cell temperature conditions of the main calibration, mole fraction enhancements (above the background) of at least 2.3 ppm would be required for the temperature effects on gain factor to be larger than the instrumental noise (characterised by

the 10 Hz Allan deviation). Such an enhancement is relatively large compared with typical ambient sampling of plume emissions at a distance from a source.

In addition to these specific tests, we utilised sampling from the 19-hour MGGA Allan variance test, where variability in cell temperature and pressure was characterised. This is also now described in the supplement. Though the cell pressure and temperature variability ranges were limited to a "reasonable" range of typical ambient conditions, these parameters showed no obvious methane mole fraction correlation, with Pearson's correlation coefficients of -0.3835 and 0.4849, respectively. Unfortunately, housekeeping data were not recorded by the pMGGA during the Allan variance test, but in principle, any correlation should be similar as the same spectroscopic fitting is used in both instruments. These results are now summarised in the final paragraph of section 2.1.

*2. What's the reason behind the exponential decay of H2O with CH4 mole fraction? Is it due to line interferences? It is difficult for readers to judge when the wavelengths of H2O, CH4 are not given. Is there an interference between CO2 and H2O as well? Notice that the exponential fits in Figures S3&S4 are based on very limited data points. What's the air matrix of the 100 ppm CH4 cylinder? Could the dependence of H2O measurements be caused by other species?*

We realise that the choice of exponential fit is arbitrary, but we chose this as it provided a suitable fit the available data points. However we also now note that that the WMO calibration scale only extends up to 5 ppm of methane, so attempting to model the water baseline any higher than this is problematic and potentially meaningless. We have therefore changed our approach and we now instead use a linear fit across the methane mole fraction range.

There is a difference between the water and methane absorption peaks of 0.2 nm which is now stated in paragraph 1 of section 2.1. This line separation is sufficient that we do not expect spectral line overlap to have a significant impact here, as the reviewer rightly points out.

All cylinders used during the water correction contained a synthetic air mix that included ambient levels of argon. While including argon in the mix is important for the calibrations (as it ensures the dry air broadening for the cylinder measurements is representative of dry air broadening for the sample measurement) we do not expect the air matrix to affect the water baseline because this represents the retrieved water vapour mole fraction for a dry sample. Therefore changes in the water baseline reflect changes in the baseline offset, not the line broadening, for this absorption line.

*3. Regarding the uncertainties of the estimated fluxes σF, what are the fractional contributions due to individual components? This information may help reduce the uncertainties in future measurements.*

This is a very useful idea. The $\sigma_F$ uncertainties were derived from flux density uncertainties, which were in turn derived by combining individual statistical uncertainties in quadrature. Therefore fractional uncertainties (that sum up to form $\sigma_F$) cannot be derived. However it is possible to derive $\sigma_F$ for individual uncertainty components, assuming other uncertainties to be zero. This revealed that uncertainty in wind speed was the dominant source of uncertainty. This conclusion is now given in the paragraph 3 of section 4, with individual $\sigma_F$ results given in the supplement.

*Detailed comments:*

*L92-93: The flow rate needs to be given when the e-folding time is discussed. Alternatively, the e-folding volume can be provided.*

This is a very useful comment. The flow rate through both instruments is now given in Table 1.

*L97-98: The unit should be ppb instead of ppm.*

Thank you for spotting this. This has now been corrected.

*L159-162: should make it clear that +0.27% and +1.8% are the differences between with and without the water vapour corrections, instead of an increase of measurement accuracy.*

The suggested change has now been made.

*L207: Equation 10 should use the molar density of dry air since CH4 is given in dry mole fraction.*

The reviewer raises a very valuable point here. We realise that we have thus far used wet methane mole fraction when this should have been converted into dry mole fraction. This requires the humidity to be taken into account when calculating molar density. We have renamed the "molar density of air" as "molar density of dry air" in paragraph 2 of section 2.4 to reflect this. We are thankful to the reviewer for raising this.

*L234-239: what was the nominal flow rate? Was the flow rate recorded? What fraction of measurements on average were omitted from each flight?*

Although the flow rate was not recorded, the average volumetric flow rate through the tubing can be calculated to be $(110\pm10)$ cm$^3$ s$^{-1}$ from the volume of the tube and the measured lag time, which was determined as the time between a spike (cough) at the inlet and the instrument response. This has been added to paragraph 3 of section 3.3 alongside the measured flow rate through the instrument of $(27.90\pm0.05)$ cm$^3$ s$^{-1}$.

16% of all sampling conducted by UAV1 was omitted due to kinks in the tubing. This now stated paragraph 3 of section 3.3.

*L239-240: what was the flow rate through the pMGGA?*

The flow rate was measured to be $(5.08\pm0.02)$ cm$^3$ s$^{-1}$. This has been added to Table 1 and to paragraph 3 of section 3.3.

*L326-330: Comparing T1.1 with T1.2 in Figure 3, I expect that larger emissions would be quantified for T1.1 and with larger uncertainties, however, the results showed the opposite. Why is that? Where are the centers of the plumes found?*

The reviewer makes an interesting observation. While it may appear as though survey T1.1 should derive higher flux as there were more plume intersections, on close examination of where T1.2 intersected the plume near the ground, methane mole fraction enhancements in T1.2 were actually far higher than in T1.1. This can be seen more clearly in the time series in Figure 2, where mole fractions of up to 12.5 ppm were recorded for T1.1, whereas up to

25.3 ppm was recorded for T1.2. Both plumes were horizontally centred near the centre of the sampling plane, with T1.1 centred at -3 m and T1.2 centred at -5 cm from the centre line, according to the NGI method. The importance of taking into account the magnitude of flux density peaks, as well as the number of plume intersections is now described in a new paragraph (paragraph 4) of section 4.

*L335-341: It looks that the crosswind distance is not sufficient to cover the plume, especially for UAV1. Why were the transects of UAV1 not centered? The current sampling tends to miss the center of the plume.*

There may be some misunderstanding here, as there may be some ambiguity in the description of our flux quantification methodology, which we hope we have now improved. The sampling shown in Figure 3 and Figure 4 may appear as if the time-averaged plume was not centred. This was because the time-invariant plume was narrow (when nearer the source) and appeared episodically due to coincidental intersection with the UAV. When nearer to the source (approximately 50 m away), the time-invariant plume can be conceived to have had less time to disperse compared to sampling approximately 100 m away. Therefore it may appear as though the time-averaged plume was not centred in our sampling plane. We now explain this in the final paragraph of section 3.3. However, in reality, where the time-averaged plume is centred on our arbitrary sampling plane is irrelevant to the flux calculation, so long as the time-averaged plume morphology can be characterised (as is the case here).

During our sampling campaign, we ensured that we flew on a downwind sampling plane, with the UAV approximately centred downwind of the source using wind direction measurements. The flight strategy was decided on each day based on wind forecasts and on-site surface wind measurements to optimise downwind sampling of the turbulently advected emission plume. This point is now made in paragraph 2 of section 3.1 and its implications are discussed in paragraph 4 of section 3.3.

---

## Author Comment (AC2) · 6 Dec 2019

We thank the reviewer for taking the time to review our manuscript and for some very useful and constructive comments and suggestions. We have responded (red text) to each comment (black italic text) in turn below. We have used these comments and suggestions to update and improve a revised version of the manuscript.

*Review*

*Testing the near-field Gaussian plume inversion flux quantification technique using unmanned aerial vehicle sampling*

*Adil Shah et al.*

*The authors use a near-field Gaussian plume inversion (NGI) technique to determine the strength of methane emission sources. In that context, they describe, characterize and deploy two commercial near-IR off-axis integrated cavity output spectroscopy (ICOS) spectrometers from ABB/Los Gatos Research. One instrument is ground-based and samples through a tube from the UAV, while the other, more lightweight instrument is deployed onboard the UAV. The authors rightly claim that more reliable techniques are needed to determine emissions from diffuse methane sources. The paper has two main technical sections: chapter 2 "Methane instrumentation and calibration" and chapter 3 "Method Testing". After struggling hard with chapter 2, I decided not to continue the review, and I suggest that the paper must be fundamentally rewritten, with respect to both language and content, before publication in AMT.*

We regret that the reviewer has identified a number of issues with section 2. We have worked hard to address all of the specific points raised by the reviewer by making significant modifications to section 2 of the revised manuscript We hope that this improves its readability.

*Chapter 2 has four subsections:*

*2.1 Instrument inter-comparison (20 lines) Jonas, please check # lines*

*2.2 Water vapour correction (61 lines)*

*2.3 Calibration (28 lines)*

*2.4 Methane enhancement and uncertainty (20 lines)*

*Section 2.1 d contains very little actual intercomparison, and thus the title is misleading The inter-comparison consists mainly of one table containing the manufacturer's specifications and the key numbers of one Allan Variance plot (based on laboratory measurements), which is given in S1. In the context of this paper, which attempts to use the instruments under harsh field conditions, much work could and should have been done to provide useful, real-world characterization under field conditions, beyond what the manufacturer reports. This may include (but should not be limited to) performance characteristics and comparison (see section title) of long-term stability, stability of calibration factors, temperature and pressure dependence, sensitivity to vibration (UAV), linearity, selectivity, (measured) response time, concentration range, etc.*

We agree that the title of this section may be misleading. We have therefore renamed the section title, "Instrumental overview", to reflect the more general scope of the comparisons

made in this section. As the reviewer rightly points out, the Allan variance plots are useful and have now been moved to the main manuscript.

Unfortunately, we lacked the facilities to replicate some UAV sampling conditions, such as UAV vibration, in the laboratory. However, we can see no reason why vibrations this should affect measurements, especially for the MGGA which was stationary on the ground

 We disagree that we sampled in harsh conditions for this work. We sampled in the bottom 50 m of the planetary boundary layer, where pressure and temperature changes are relatively small compared to laboratory conditions, especially in temperate summer months. To better convey this, the sampling conditions have now been plotted alongside the calibration conditions, for both instruments, in the supplement. Furthermore, as suggested, we now discuss the very small potential effect of temperature and pressure variations on measured methane mole fraction for the MGGA in the final paragraph of section 2.1, which showed no discernible impact on measured methane mole fraction with changes in the ambient range of cell temperature and cell pressure during the Allan variance test.

We believe that the temporal stability of calibration factors was already captured in section 2.4, where the standard deviation in offset and gain factor is given over a long calibration period in the laboratory. However the reviewer rightly points out that we failed to capture the uncertainty due to linearity which we have now determined and included in the revised manuscript in the final paragraph of section 2.4.

The measured response time depends on the length of the inlet, as air takes time to travel through the external and internal tubing. This was measured in the field. Therefore these details are provided in paragraph 3 of section 3.1 where it is more relevant to the specific instrumental set-up.

*Section 2.2 is the core part of the instrumental section, which is reflected by its length. It covers an in-depth description of water vapor correction but sadly contains a number of fundamental flaws. Some examples are below, but this list is not exhaustive:*

*- The largest correction to obtain dry $CH_4$ values ($[X]^{dry}$), needed for flux calculations, is water vapor. While the authors do many tests and calculations (see below) to get this value, they never show how accurate the $H_2O$ measurements of the instruments are. They likely have the information (based on the measurements with a dew point generator), but the corresponding performance is never shown. For such a detailed water vapor study, it would be key to deduce this effect first and then discuss (and correct) the remaining influencing parameters in a second step.*

The reviewer rightly points out that we do not accurately derive water vapour mole fraction in the cell. As we conduct an empirical water correction, which by its nature is based on reported methane mole fraction, it is not necessary to derive water mole fraction accurately, based on the conclusions of Rella *et al* (2013). However this empirical correction requires the water vapour mole fraction measurement to be stable and independent of methane mole fraction. As it was observed that the water baseline was in fact affected by methane mole fraction, our water baseline corrected for this systematic effect. In order to emphasise the fact that our water correction is purely empirical and is not based on spectroscopic first principles, we have renamed the section "Empirical water vapour correction" and we have redefined *v* as an "empirical water correction factor". To be clear, we do not attempt to characterise the suitability of these instruments for measurement of water vapour concentration, which would

require a separate evaluation focussed on water vapour with a different (high accuracy) water reference instrument. This may constitute future work, but the scope of this paper is on methane measurement only, with water vapour as a "problem" to be accounted for.

*- Figure S3 shows the effect of $CH_4$ on apparent water, due to cross sensitivity of the analyzers. Equation 1 has no physical meaning or motivation, but it produces an exponential curve, which fits the data points nicely. If there is no clear motivation, such a weak set of data may as well be described by a linear interpolation (see S3). Furthermore, applying the same parameter ($\omega$) for the other analyzer, which shows clearly different absolute and relative behavior (S4), is not justified.*

The reviewer is correct to point out that the exponential baseline fit was arbitrary and was used to fit the data well. On reflection, we realise that the WMO calibration scale only extends up to 5 ppm, so any water baseline correction above 5 ppm is meaningless, without an equally good test for linearity up to a higher mole fraction using certified standards. To this end, we agree with the reviewer and have now applied a linear regression to the baseline from both instruments, over the range of the WMO scale.

As we observed the water baseline to behave non-linearly at 100 ppm methane mole fraction in the MGGA, we have also applied a linear fit including the additional 100 ppm data. This allowed us to assess the change in mole fraction measurements with the 100 ppm data included in the baseline to be, on average, (0.02±0.03)% higher for all MGGA sampling recorded from UAV1. This analysis is now presented in the supplement.

*- Somewhat similar is the issue in S5 and S7. It is not surprising to find a (nearly linear) dependence between v and water vapor. It is also not surprising that the residuals become a bit smaller if using a second order polynomial. However, since there is no information on the long-term stability of this system (e.g. repeatability and reproducibility), these fit characteristics are of very little use (a larger order polynomial would give even smaller residuals).*

The choice of a second order polynomial fit was based on polynomial water corrections used in previous work, using similar spectroscopic techniques (O'Shea *et al.* (2013), for example). Furthermore, O'Shea *et al.* (2013) (who used the same off-axis ICOS technique but a different instrument) found that v remains consistent over time with negligible temporal drift. This valuable point has now been made in the final paragraph of section 2.2.

*- The authors call v the water correction factor. This, again, is misleading because it contains normalization to dry conditions, which is a standard procedure, and corrections for non-perfect spectroscopy (which is specific to their analyzers).*

We realise that this may be misleading and have therefore renamed it an "empirical water correction factor".

*- Based on these badly justified fits and corrections, the authors deduce a water uncertainty factor ($\sigma_v$). However, $\sigma_v$ is the standard deviation of the mean of the residuals in S6/7. This parameter carries information about the quality of the fit of the polynomial, but no information about the quality of the water correction for a specific measurement at a given $CH_4$ and $H_2O$ mole fraction.*

The reviewer rightly points out that our uncertainty is based on the quality of our polynomial fit. We simply cannot derive an uncertainty on something that is already uncertain. We have to assume that the uncertainty is represented by the goodness of the fit we can produce from empirical data. We have used the same approach as Rella *et al.* (2013) and O'Shea *et al.* (2013), as guidance of optimal practice. We have therefore clarified this in the final paragraph of section 2.2.

*Sections 2.3 and 2.4 describe the calibration procedure and derive measurement uncertainty.*

*- Calibrated enhancement in methane mass density (E) is calculated based on parameters that are obtained in the above experimental and fit procedures. The procedures themselves are correct (but see some remarks above and below). The authors then state that their approach is especially useful because it can be used to determine measurement uncertainty (equation 11). However, this uncertainty calculation is based on laboratory measurements and badly motivated fit procedures. It does not contain non-linearity nor any changes (drifts) in time, e.g. of G and v. Furthermore, it does not include any field (in)stability considerations, nor the fact that one parameter is just assumed to be the same in MGGA and pMGGA. A realistic uncertainty budget would be highly welcome, but the approach chosen here is useless.*

We thank the reviewer for highlighting the fact that we had not previously accounted for non-linearity. This has now been quantified up to 5 ppm by introducing a new non-linearity uncertainty parameter ($\sigma_L$), which has been incorporated into the enhancement uncertainty, as suggested.

With regards to drifts, the variability in $G$ was already included in the enhancement uncertainty, represented by $\sigma_G$. As $\sigma_G$ was derived from changes in $G$ over time, we believe that it implicitly captures temporal drifts. We have now made this clear in the second paragraph of section 2.4. Additionally, we assume here that $v$ does not drift, based on previous work using a similar instrument described by O'Shea *et al.* (2013).

*Some less fundamental comments are below, illustrating that the whole paper should be strongly re-vised:*

We have addressed all of the reviewer's comments and we have significantly revised the manuscript to take these comments into account.

Abstract

*Page 1 – Line 12*

*Many parameters contribute to the uncertainty. Are they large or small, and is the contribution due to the "maybe" poorly quantified sources important or neglectable?*

This sentence simply motivates the background to the paper. We are not tackling the global methane budget problem head on in the work. The choice of the words "may be" is due to the large uncertainties within the source and sink terms in global methane budget and the widely recognised difficulty in attributing the cause of these uncertainties. Section 1 expands on this and cites relevant papers that discuss the global methane problem. To make this clearer for the purposes of our manuscript and the abstract, we have added the words "potentially" and "many", so the sentence reads, "Methane emission fluxes from many facility-scale sources

may be poorly quantified, potentially leading to uncertainties in the global methane budget". We hope this clarifies our point of view.

*Page 1 – Line 14*

*This is not a NEW near-field Gaussian plume inversion (NGI) technique – it has been published before by the same authors.*

The word "new" has been removed. We instead claim that this is the "first test" of the NGI method using "unbiased sampling".

*Page 1 – Line 20*

*Modified with respect to what?*

This is a good point. It was not previously clear what this was with respect to. The sentence has been rephrased to make it clear that the water correction was specially adapted for the instrument. The sentence now reads, "a water vapour correction factor, specifically calculated for the instrument, was applied and is described here in detail".

*Page 1 - Line 23*

*Simplify sentence, e.g. the uncertainty was between ... and ...*

We appreciate that this sentence can be simplified. However the quoted values are not uncertainties but rather uncertainty bounds as a fraction of the controlled emission flux. We have rephrased the sentence to simplify it, whilst maintaining the correct meaning.

*Page 1 – Line 25*

*The term "range" implies that it is the range between 17 and 218 %, but this is not meant here.*

The term "These highly conservative uncertainty ranges" has been replaced by "This range of highly conservative uncertainty bounds". This is now less ambiguous.

*Page 1 – Line 27*

*"flux approaches": check language*

The term "flux approaches" has been replaced by "flux quantification approaches".

*Page 1 – Line 27*

*Replace "may perform well" by "may be a valuable alternative"*

This replacement has been made.

*Page 1 – Line 28*

*"applied to UAV sampling" should read "combined with UAV sampling"*

This replacement has been made.

**Introduction**

*Page 2 – Line 53*

*precision should read accuracy or precision and accuracy*

The suggested change has been made.

*Page 2 – Line 54*

*have not yet*

This replacement has been made.

*Page 2 – Line 60*

*This suggests that controlled release is the only acceptable method. Make statement more general, e.g. the method was not validated, and its uncertainty was not quantified (e.g. with controlled release experiments).*

Although some methods can be mathematically correct on paper and conceptually sound, we stand by the idea that any method should be tested by sampling emissions of an accurate known flux rate to prove its utility. However we acknowledge the reviewer's point that a controlled release from a gas cylinder is not necessarily required and other approaches could be adopted, so long as a flux is known. We have rephrased the sentence so it reads, "However this method was not tested for UAV sampling with an accurately known (controlled) methane flux rate", to make it more general and allow for testing using any known flux.

*Page 2 – Line 62*

*Make it clear that "Shah et al" is not any other researchers but the same group, e.g. our previous study...*

The suggested change has been made.

*Page 2 – Line 65*

*Language: the sampling does not develop the technique. e.g.: ... sampling ... used in combination with ...*

The downwind sampling used during this previous study was used to develop our flux quantification technique. The technique did not exist before the sampling campaign. The measurements we acquired showed us that a new flux quantification approach was required, as previous approaches failed. We have now added this point to the sentence, to make it clearer that the new approach was a consequence of the failure of other approaches. We have also clarified that the "data-set" from the previous sampling campaign was used to develop the new method.

**Methane Instrumentation and Calibration**

*Page 3 – Line 97*

*$\sigma_{AV}$ is a very unusual parameter. Contains very little information because it depends on averaging time. "Allan deviation uncertainty factor" is confusing.*

We realise that the Allan deviation at the maximum sampling frequency ($\sigma_{AV}$) is not so useful for direct instrument comparison, which is why we also provide the Allan deviation at the same frequency (both 1 Hz and 0.1 Hz) in Table 1. For our purposes, as both instruments sampled at their maximum sampling frequency (or minimum averaging time) and these were different for each instrument, $\sigma_{AV}$ is used to quantify instrumental noise (for the instrumental averaging time used during sampling) and serves as a component of the total mole fraction enhancement uncertainty. We have renamed $\sigma_{AV}$ as "sampling noise uncertainty" to make this less ambiguous.

*Page 3 – Line 108*

*"These three effects have a net effect of decreasing [X]$_0$ in both instruments". You probably mean "decreasing [X]". Furthermore, this is only useful information if you state before which effects have a positive/negative impact. Whether the sum is a net negative effect depends on the conditions (usually dilution dominates, so the statement is not wrong).*

We feel there may be some misunderstanding here in terms of the effect of water vapour on [X]. [X] is a true mole fraction measurement (*i.e.* the true dry mole fraction in the atmosphere), independent of the instrument, whereas [X]$_0$ is an instrumental reading which can be different to the true value because it may be affected by conditions such as atmospheric water vapour and is not calibrated.

Regarding isolating the individual water vapour effects on [X]$_0$, we do not deem it empirically possible as what the instrument observes is decreasing [X]$_0$ with increasing water mole fraction. We have described all three effects in the manuscript to provide a background and motivation for our correction procedure. We believe it is unnecessary to fully characterise each of the three individual effects if the combined effect can be adequately characterised empirically. We have now clarified that we observe consistent decrease in [X]$_0$ at a range of methane and water mole fractions.

*Page 3 – Line 110*

*If the pressure broadening coefficient should account for effects two AND three, then state so explicitly.*

We have now clarified this in the first paragraph of section 2.2 that spectral overlap is expected to be an issue. Therefore we have now made it clear that the algorithm only corrects for pressure broadening in paragraph 2 of section 2.2.

*Page 4 – Line 122*

*Has a calibration for water vapor been performed, and what are the results?*

This is addressed in our response to the comment regarding Section 2.2 above. In summary, we do not characterise this instrument for water vapour measurement.

*Page 4 – Line 132*

*"more points" means: 1.9, 2.1, 5.0 and ca. 104 ppm. It does not make sense to derive parameters of a (complicated) exponential function if there is no physical concept that supports this function.*

We agree and have replaced it with a linear regression, over the range of the WMO scale (up to 5 ppm).

*This $H_2O$ baseline is very large; your data in S3 suggests that 100 ppm (dry) $CH_4$ results in ca. -5500 apparent ppm $H_2O$. Is this analyzer malfunctioning and should be returned to the manufacturer, or is it within the specifications for water vapor measurements? Using the MGGA $H_2O$ baseline function "as-is" for another analyzer does not make sense, especially since the experimental values for the pMGGA (S4) are significantly different from those of the MGGA, indicating that the two functions may well not be the same. Alternatively, one may just acknowledge this fact and do a corresponding uncertainty budget. In the context of the large uncertainty of flux estimation with NGI, this may be appropriate. However, it would fundamentally change the arguments and line of thought in this paper.*

We do not evaluate the performance of this instrument for water vapour measurement, which is beyond the scope of the manuscript. In this work, we are interested in the effect of water vapour on our target gas, methane. Water vapour influence is therefore something we simply seek to account for. However, the instrument specifications suggest that the instrument is designed to sample up to 0.03 $mol_{water}$ $mol^{-1}$, with a manufacturer specified accuracy of 0.2 $mmol_{water}$ $mole^{-1}$ at 1 Hz. The objective of characterising a water baseline was to ensure that the subsequent water correction was independent of the effects of methane in the cavity.

Regarding our choice of baseline, we agree with the reviewer and now use fully independent linear baselines for each individual instrument.

Page 4 – Line 135

"Having established a well characterised water baseline…"

This baseline may be well characterized. However, it corresponds to a very large correction, which can-not easily be redone in the field. Therefore, it is necessary to measure the longterm stability of the baseline and instrument-to-instrument variability. Otherwise the "well characterized" is not helpful.

This is a very useful point. We are confident that that the water baseline is relatively stable. To test this, we also conducted an Allan variance test on the dry water baseline for both instruments, at 2 ppm of methane. This revealed that the Allan variance of the MGGA and pMGGA (at the averaging time used to derive the baseline) was $\pm 16 \cdot 10^{-6}$ $mol_{water}$ $mol^{-1}$ and $\pm 27 \cdot 10^{-6}$ $mol_{water}$ $mol^{-1}$, respectively. This information is now added to paragraph 4 of section 2.2. These water baseline values are far smaller than water mole fractions sampled during the water correction procedure of up to 20 000 $\cdot 10^{-6}$ $mol_{water}$ $mol^{-1}$, also at 2 ppm methane.

*Page 4 – Line 136*

*What was the mole fraction of $CH_4$ in this cylinder?*

The methane mole fraction was 2.205 ppm for the MGGA and 2.183 ppm for the pMGGA (different cylinders were used). This is now stated in paragraph 4 of section 2.2.

*The $H_2O$ baseline has a huge dependence on $CH_4$. Nevertheless, this experiment was only done at one $CH_4$ concentration only. This may not be representative and add uncertainty to the retrieval of the "real-world" measurements.*

The concentration of methane in the gas cylinder was 2.2 ppm, which is similar to the current atmospheric background of approximately 1.9 ppm (depending on the time of year). Thus the water correction was performed in the most real-word conditions possible, to minimise uncertainty.

Furthermore, pressure broadening line shape changes and dilution affect the instrument methane gain factor, with increasing water mole fraction. There is no physical (instrumental) reason for these effects to manifest themselves as an effect on the instrumental methane offset. Therefore, a water correction derived at one methane mole fraction should be equally valid at other methane mole fractions.

*Page 4 – Line 150*

*The difference between wet and dry corresponds to humidity, thus typically 1-3 %. This is not "almost equal". If it has little impact on the result, then this is because the retrieved apparent $H_2O$ does not very strongly depend on $CH_4$ under the conditions used to test the simplification.*

We accept that the phrase "almost equal to" may be misleading. We have therefore rephrased this as "close to". We have also now clarified that the water correction has little overall impact on dry mole fractions in the penultimate paragraph of section 2.2, in our simplified test, where it is referred to as a "simple example".

*Page 5 – Line 156-157*

*The uncertainty of water measurements cannot be derived using "the water correction residual (R) from Eq. (2)". The calculated value is not the uncertainty of water measurements, nor the uncertainty after water correction. It is the standard deviation of the mean of the residuals. This parameter carries information about the quality of the fit of the polynomial, but no information about the quality of the water correction for a specific measurement at a given CH4 and HO mole fraction.*

The reviewer correctly points out that we do not present an uncertainty in the water correction factor, but instead an uncertainty in our fit. This was also addressed in our response above, where this was raised. We now make this clear in the final paragraph of section 2.2. Furthermore, the water "uncertainty factor" has been renamed the water "fitting uncertainty factor" to reflect this.

*Page 5 – Line 160*

*"For example our correction was used to increase 160 the MGGA measurement accuracy of $[X]_0$ (at 2 ppm) by +0.27%, at a humidity of 0.001 mol water $mol^{-1}$, and by +1.8%, at a humidity of 0.01 mol water $mol^{-1}$". This is a meaningless statement. It does not consider that any correct measurement includes the correction of the dilution effect. You compare apples with rotten pears.*

This sentence may have been poorly written and misleading. The purpose of this sentence is to convey that we can improve the measured mole fraction, *i.e.* it more closely agrees with

the true dry mole fraction. We have replaced the phrase "increase the MGGA measurement accuracy of $[X]_0$" with "increase $[X]_0$". This reduces ambiguity, to make it clear what the correction achieves.

*Calibration*

*Page 5 – General Comments on Section/Calibration*

*Useful:*

*The alternating measurement of two standards. This gives the repeatability of measurements under laboratory conditions. A simple average and standard deviation would characterize this. Alternatively (x - average x) vs. N (in bins) as a figure would illustrate that the values show a normal distribution.*

*Not useful:*

*The concept of uncertainty in C and G because it neglects the fact that the linearity in response is most frequently the key factor determining the uncertainty of the measurement.*

*Needed:*

*Determination of linearity.*

The reviewer raises a valuable point regarding quantifying uncertainty due to non-linearity. During our main calibrations, only two standards were used (2 ppm and 5 ppm), as the World Meteorological Organisation Greenhouse Gas Calibration Scale does not extend higher than 5 ppm and reliable alternative standards cannot readily be obtained. Therefore, we have now tested the linearity within the calibration range, by sampling 5 gases with the MGGA. This showed that the residual uncertainty between measured and certified $[X]$ was on average 2.3 ppb. As we no longer have access to the pMGGA, we assumed similar linearity in both instruments as they use identical spectroscopic techniques. This test is described in section S4 of the supplement. The non-linearity uncertainty has been incorporated into the overall enhancement uncertainty.

*Page 5 – Line 168-169*

*This does not make sense. Water vapor in reference gas cylinders should be very low if filled dry (which I assume was done here). Even if filled wet, then 60 bar/20°C should be significantly below 500 ppm. Thus, $H_2O$ from a pressurized bottle (if treated correctly) is so low that it has very little impact on the measurements. An additional trap (as used here) may well lead to water contamination if one is not very careful.*

We used a water trap as these cylinders were synthetic blends, prepared to include ambient levels of argon to provide an appropriate air matrix. This may cause a small amount of water to be present in the cylinders.

*Page 6 – Line 195-196*

*C deviates by about 12 ppb from 0. Whether that is good or bad (and under what conditions) depends on the application. However, to say that this is "almost equal to 0" for an analyzer with an Allan Minimum below 1 ppb is bizarre.*

This is a valid point. 12 ppb is indeed much larger than a 3 ppb Allan deviation (at 10 Hz) for the MGGA. However when compared to the atmospheric methane background of approximately 1800 ppb, 12 ppb is very small. The sentence now includes this qualification, to clarify our point of view.

*Page 6 – Line 195*

*$v$ is ca. 0.98 at 1 % H2O. This is not "almost equal to 1". Furthermore, v in S6 contains all water-related corrections, including normalization to dry conditions. Stating that this is "almost equal to 1" does not make sense, because the correction to dry conditions should be done in any case.*

We recognise the reviewer's point. We have therefore removed any reference to $v$ from this sentence.

*Additional remarks*

*"A lightweight wind sensor (FT205EV, FT Technologies Limited) was mounted on-board UAV1, on a carbon fibre pole 305 mm above the plane of the propellers. It recorded wind speed and direction at 4 Hz. This data was used to model the change in wind speed with height above ground level (z)."*

*- Was this sensor validated?*

The wind sensor was used to derive a wind-height profile, for cross comparison against stationary ground-based wind measurements. The key need here was a linear response to wind speed changes. This was tested by comparing wind speed at 3.3 m from the UAV sensor to a stationary 3.3 m anemometer. A linear fit intersecting the origin agreed with all points, within uncertainty, implying linear response of the UAV wind sensor. These results are now presented in section S7 of the supplement.

*- Are the measurements independent of the propellers? How was this validated?*

The effect of air disturbance on the measured wind field due to rotating propellers was minimised as the wind sensor was positioned 0.3 m above the plane of the propellers. Unfortunately, we did not have required facilities (such as a wind tunnel) to fully test that this was the case. However each UAV transect was carefully planned to minimise this effect by rotating the UAV orientation at each turning point. This would effectively cancel out any potential residual net wind vector distorting the wind field due to the rotation of the propellers. This is now stated in section S7 of the supplement.

*- How reliable were the direction and speed when the drone was moving? How was this determined?*

The reviewer is correct to suggest that the UAV wind measurement may have been affected due to the movement of the UAV. This impacts the measured wind vector component parallel to the direction of movement of the UAV. Therefore only the wind component perpendicular to the orientation of the UAV sampling plane was used. The motion of the UAV has no impact on the wind component perpendicular to the plane of movement of the UAV. It is the perpendicular wind component that is used to construct the wind profile used to derive flux. This rationale is now explicitly stated in paragraph 1 of section S7 in the supplement.

*One analyzer was using an inlet just underneath the drone. The corresponding sample is representative of the air above the drone because of downwash. Was this considered? How?*

This is an astute comment. The reviewer is right to point out that mixing due to downwash could occur and that what is sampled is air from a subtly higher elevation to the position of the inlet. The inlet was 31 cm below the plane of the propellers. We have not (and cannot usefully) account for this small height difference in the calculation of flux as the air sampled is likely mixed in a thin (31 cm) layer between the plane of the rotors and the plane of sampling.

*Flux of one drone --> the other just sampled further away.*

We do not understand what the comment above is suggesting.

*The Allan Variance plot should be in main section of the paper. Its v-shape is relevant for interpreting the measurements during flight (i.e. 10 – 30 minutes, typically).*

Both plots have now been moved to the main manuscript, as suggested.

---

## Author Response (AR2)

Dear Editor,

We thank the associate editor for their timely response and for their useful advice. We have addressed the issues raised by reviewer 3 to the best of our ability except, where the associate editor rightly highlights, improvements were not logistically feasible due to return of the pMGGA instrument or lack of access to additional gas standards.

To address the limitations of the sensors, we now clearly state in the abstract that the sensor characterisation is suitable for our UAV sampling purposes but that temperature and pressure effects have not yet been comprehensively characterised. Temperature and pressure characterisation is presented as a topic of future work. We have also added a sub-section to the end of section 2, summarising potential future improvements for sensor characterisation. This now outlines ways to improve sensor characterisation, either by running calibrations in controlled environments or by sampling multiple gas standards. We also further discuss sources of (very small) systematic uncertainty associated with an assumption of linearity in the water baseline, in response to reviewer 3.

We have compiled our point-by-point response to comments made by both reviewers below. We have also included a marked-up version of the modified manuscript, addressing all of the reviewer's concerns.

Yours faithfully,

Adil Shah

**Response to reviewer 1**

*The reviewer appreciates the extensive responses from the authors. After reading through the responses to both reviewers' comments, my impression is that it may be Okay to use the two instruments MGGA and pMGGA to measure large signals as in the CH4 release experiment, but the uncertainties of the two analyzers have not been thoroughly/convincingly characterised, which unfortunately limits the potential use of such instruments in other applications. This is especially the case when a direct comparison of such an instrument with a proven technique is lacking.*

We thank the reviewer for taking the time to review our manuscript for a second time. We also thank the reviewer for acknowledging our improvements. We regret that the reviewer still deems our additional analyses to be insufficient, in light of the fact that UAV sampling conditions (see Table 5) were similar to conditions during laboratory testing. However we recognise that further future characterisation by others may well serve to improve measurement accuracy. Indeed, characterisation of any instrument is something that should always be done and is all too often overlooked by some. We have therefore now stated clearly in the abstract that temperature and pressure effects were suitably characterised for our UAV sampling experiments, but that future characterisation may be necessary depending on the application of measurements. We appreciate that the constraints of our characterisation work mean that they may not be applicable for use in more extreme environments. To highlight recommended future sensor characterisation improvements, we have added a new sub-section to the end of section 2 summarising these ideas (calibrations in a controlled environment and sampling additional gas standards). We hope these additional modifications are more acceptable and appropriate in light of the principal (UAV sampling) focus of the manuscript.

**Response to reviewer 3**

We thank the reviewer for taking the time to review our manuscript and for some very useful and constructive comments and suggestions. We have responded (red text) to each comment (black italic text) in turn below. We have used these comments and suggestions to update and improve a revised version of the manuscript.

*General Comments:*

*1) In S3 and S4 you derived the [H2O] offset by measuring two or three different concentrations. I think two points alone does not make a strong enough correlation for a calibration curve (typically you need at least 4-5 for this practice). This however has created a very large spread on the ends of your distribution, and it doesn't necessarily tell you much about if the behavior is linear in between the points. For example, in S3, your point between the extremes lies under your line of best fit. So perhaps it would be best to present your endpoints as points with error bars to better represent the uncertainty. If possible, I would recommend running the pMGGA with the 2.167 cylinder and both with one more standard. You may also consider adding a blank (0 ppm), as is common in most spectroscopic practices. However, I think the method you used of drying the gas is acceptable. I also agree with the assertion that this will stay relatively constant over time.*

*The same thinking goes to S6 and S7. This treatment may help determine if the exponential truly is the line of best fit in S9 and S10.*

The reviewer makes a valid comment regarding our choice of water baseline fit and this is an aspect we addressed in a previous manuscript discussion. The use of a linear fit was actually recommended by previous reviewers. We have conducted further work to assess the systematic uncertainty associated with the use of such a baseline fit (see below). We agree with the reviewer that three data points is insufficient for a complete understanding of the potential linearity of the water baseline. We also agree that it would be better to sample more gas standards. Unfortunately, more gas standards are not available to us at present. Furthermore, the prototype instrument has now been returned to the manufacturer so we cannot conduct further characterisation work. However, as the water baseline is a very small component of the water correction factor, our assumption of linearity is sufficient, given the large mole fraction enhancement uncertainty terms in equation 11, including $\sigma_L$ of ±2.3 ppb and $\sigma_n$ of at least ±2.7 ppb. To verify this assumption, a sensitivity test was conducted, assuming the water baseline at 2 ppm to in fact be the water baseline 5.1 ppm. In principle, this would erroneously reduce mole fraction (at 2 ppm) by only 2 ppb. Thus even if our assumption of non-linearity is utterly false and there is no trend at all, the increase in systematic uncertainty would be negligible when compared with instrumental precision. This sensitivity test is now described and discussed in paragraph four in section S2 of the supplementary material.

We thank the reviewer for highlighting the spread in water baseline values. We have now added a standard deviation bar alongside each individual data point, for each gas cylinder, in Figure S3 and Figure S4 as suggested. This now better highlights the variability in water baseline and better captures the magnitude of uncertainty in our fit.

The reviewer has made an astute observation that one of the three sampled gas cylinders in Figure S3 (for the MGGA water baseline) resulted in a water baseline below the general linear trend. Having investigated this further, we now realise that this was most likely due to slightly different environmental conditions when sampling this gas cylinder compared to the other two (now described in paragraph 4 of section S2 of the supplement). Instrumental housekeeping data revealed that the cell pressure was 9.4 mbar higher and the cell temperature was 1.9° C lower when sampling this cylinder. Nevertheless, as our laboratory characterisation procedures took place under similar environmental conditions to UAV testing (see Table 5 for values), we believe any net effect on methane mole fraction to be very small and that any systematic water baseline effects is captured within the existing uncertainty terms of equation 11.

Ideally, we would have preferred to sample the 2.167 ppm cylinder using the pMGGA. However, we had limited access to this instrument, as it was returned to the manufacturer. Nevertheless, following the sensitivity test described in paragraph 1 above, the net effect of the water baseline on the overall water correction factor is small, especially for the pMGGA. Variability in water baseline for the pMGGA (between 2 ppm and 5 ppm) was $0.00005$ $mol_{water}$ $mol^{-1}$, while for the MGGA it varied by $0.0007$ $mol_{water}$ $mol^{-1}$. Thus even if the water baseline was potentially poorly characterised for the pMGGA during laboratory testing, its small variability with methane mole fraction means that any potential systematic uncertainty in water correction factor would be negligibly small compared with the inherent random error.

Testing both instruments using zero-air (a 0 ppm standard) is a good suggestion. However we did not have access to a synthetic air mix without any methane. Although we could use pure nitrogen, it would not be suitable for this test as collisional peak broadening would result in an unnatural line shape without oxygen, argon and carbon dioxide. However we acknowledge that if a suitable standard were available, this would be a very valuable test in future. This has now been suggested as a potential test for forward guidance.

In summary, we recognise that there are a number of tests that would serve to improve sensor accuracy in future, by better characterising the water baseline and deriving a gain factor across a wider range of environmental temperature and pressure conditions. These tests are beyond the scope of our manuscript as the sensors needed to be suitable for UAV sampling in ambient atmospheric conditions in the UK summer. We have confirmed that conditions were suitable by comparing the values in Table 5 to the environmental conditions during these laboratory tests. Nevertheless, tests in a controlled environment, sampling more standards, could be recommended in future work in order to get the very best out of the instrument. Therefore we have added a new sub-section at the end of section 2, highlighting these future improvements.

*2) For UAV2, where was the airflow being pulled into the pMGGA from? Did it have a boom or some other inlet that brough air into the instrument from a location undisturbed from the rotor wash? Same goes for the UAV1 and its tubing – was it located on a boom away from the propellers? Several groups have determined that sensor placement matters a great deal, so you may want to at least discuss any considerations you did make, even if this was not the aim of this work. Including a schematic of UAV 1 and 2 in section 3.1 could be beneficial to help readers see where the sensors were placed, including the wind sensor you did describe. This was very good, and I think you just need to add the inlets for the pMGGA / MGGA tubing to this section.*

The reviewer is right to highlight the importance of air inlet positioning. It has been well-documented (including our previous work: Shah *et al.*, 2019) that propeller downwash can distort the sampled spatial plume morphology. The inlet should ideally be placed above the plane of the propellers as air funnels inwards from the sides towards the centre of the UAV, with turbulent disturbance and downwash below the plane of the propellers. As UAV1 was specially designed for atmospheric sampling, both the air inlet and the on-board wind sensor were positioned on poles 0.3 m above the plane of the propellers (see Table 4 for air inlet heights with respect to the base of the UAV). A perfectly analogous setup for UAV2 was not possible, as we only had access to the pMGGA for a limited time. We now discuss these points in an additional paragraph at the end of section 3.1

We agree that including a schematic of the position of the air inlet is a very good idea. We have now added a new figure (figure 5) to the manuscript which shows photographs of both UAVs and highlights the position of the air inlet as well as the plane of the propellers, with respect to the base of the UAV.

*Detailed Comments:*

*L33-39: This paragraph does an excellent job elucidating why the methane budget is poorly constrained and how we can work to reduce our uncertainties. However, what could really put the*

*finishing touch here I believe is a sentence at the beginning highlighting the importance of the methane budget to the climate system (i.e. GHG forcing, implications on hydroxyl radicals etc.).*

We thank the reviewer for their kind comments. The reviewer makes a valuable point here. Two short sentences have been added to the start of the introduction to place the rest of the manuscript into context, as follows. "Methane is the second most important anthropogenic greenhouse gas (Etminan et al., 2016), with an important role in atmospheric chemistry processes (Ehhalt et al., 1972). There is more methane in the atmosphere today (on an average annualised basis) than there has even been over the past 800 000 years (Etheridge et al., 1998; Loulergue et al., 2008; Earth System Research Laboratory, 2020)."

*L309: Be advised the numbering in section 3 skips from 3.1 to 3.3. Please correct this to 3.2 then 3.3.*

We thank the reviewer for highlighting this error which we have now corrected.

*S17-18: If the authors are referring to the mole fraction intercept in S2, 4.3014 ppm is incorrect. The y-axis in the plot is the same in both S1 and in S2, so either S2 needs to be corrected of the text needs corrected.*

There may be some confusion here. The different figures show the (different) y-intercept for different parameters (cell temperature and cell pressure). The value we are referring to here is a y-intercept when x equals zero. This extrapolates the linear fit to the point where x equals zero as a function of cell temperature and cell pressure (separately). They are inherently different. The confusion here may simply be because the gradient "looks" similar on the two plots, whilst they each have a different intercept. To clarify this in the supplement, we have replaced "mole fraction intercept" with "mole fraction zero intercept".

**Modified manuscript**

[revised manuscript text omitted]

---

## Author Response (AR3)

Dear Editor,

We thank the associate editor for their useful technical suggestions. We have made all of the suggested changes. We have included a marked-up version of the modified manuscript, containing all of the associate editor's improvements.

Yours faithfully,

Adil Shah

Modified manuscript

[revised manuscript text omitted]